# OCCGEN: Selection of Real-world Multilingual Parallel Data Balanced in Gender within Occupations

**Marta R. Costa-jussà**
Meta AI
costajussa@meta.com

**Christine Basta**
Universitat Politècnica de Catalunya
IGSR, Alexandria University, Egypt
christine.raouf.saad.basta@upc.edu

**Oriol Domingo**
Batou XYZ
oriol@batou.xyz

**Andre Niyongabo Rubungo**[*]
Princeton University
rn3004@princeton.edu

## Abstract

This paper describes the OCCGEN toolkit, which allows extracting multilingual parallel data balanced in gender within occupations. OCCGEN can extract datasets that reflect gender diversity (beyond binary) more fairly in society to be further used to explicitly mitigate occupational gender stereotypes. We propose two use cases that extract evaluation datasets for machine translation in four high-resource languages from different linguistic families and in a low-resource African language. Our analysis of these use cases shows that translation outputs in high-resource languages tend to worsen in feminine subsets (compared to masculine), specially in the directions containing English. This is confirmed by the human evaluation. We hypothesize that a sound language generation may contribute to pay less attention to the source sentence and to overgeneralize to the most frequent gender forms.

## 1 Introduction

Biased NLP systems can mainly cause harm in allocations (e.g., giving job opportunities to particular social groups) and in representation (e.g., by propagating and amplifying stereotypes) Blodgett et al. [2020]. Typical examples are associating neutral words with one gender Bolukbasi et al. [2016] or wrongly translating the gender of an entity because of the influenced of a social stereotype Prates et al. [2020], e.g., *The doctor decided to bring her phone.* to *El\* doctor\* decidió llevar su teléfono.* instead of translating to the correct form *La doctora decidió llevar su teléfono.*.

NLP biases have several dimensions that should be tackled, including detection, evaluation, and mitigation. This paper focuses on generating balanced datasets to address evaluation and mitigation issues. Our motivation to create balanced datasets for training purposes comes from the fact that previous works have shown that fine-tuning with balanced data Saunders and Byrne [2020], Costa-jussà and de Jorge [2020] mitigates gender bias, while creating balanced datasets for evaluation aligns to further progress in responsible artificial intelligence evaluation[2]. Therefore, we propose a methodology to collect monolingual, bilingual, and multilingual datasets balanced in gender and occupations to train or evaluate Machine Translation (MT) models. The outcomes of this paper are two-fold. On the one hand, the OCCGEN toolkit can be customized according to research and development needs towards languages and gender definition (beyond binary). On the other hand, two use-cases of this toolkit are presented in this paper, which provide two evaluation benchmarks for

---

[*]Work done while at Universitat Politècnica de Catalunya

[2]https://ai.facebook.com/blog/facebooks-five-pillars-of-responsible-ai

36th Conference on Neural Information Processing Systems (NeurIPS 2022) Track on Datasets and Benchmarks.

the particular case of binary gender (masculine and feminine). One use case includes a multiparallel dataset in four high-resource languages of different linguistic families (Arabic, English, Russian, and Spanish). The other use case includes parallel dataset in a low-resource African language (Swahili) with English. We provide MT evaluation of different models with these two benchmark datasets. We analyze the impact on gender performance without requiring an additional specific measure for gender and relying on standard MT automatic evaluation methods. We additionally perform a human evaluation that evaluates the gender accuracy of marked words from our datasets. Our data[3] and toolkit[4] are freely available in Github.

## 2 Related Work and Definitions

**Related work** As follows, we summarize studies most similar to ours in terms of datasets created to evaluate gender bias. While there are proposals in a wide variety of natural language processing applications (e.g., language modeling Nadeem et al. [2021], Nangia et al. [2020]), we focus on summarizing the ones in MT. The evaluation of gender bias generally combines the proposal of new datasets and a proposed evaluation methodology. Compilation of datasets for evaluation has been approached by proposing synthetic patterns Stanovsky et al. [2019], Nangia et al. [2020], Nadeem et al. [2021] or real-world selection Costa-jussà et al. [2020], Levy et al. [2021]. Some of these datasets have been analyzed and proven to contain several critical pitfalls Blodgett et al. [2021], including unstated assumptions, ambiguities, and inconsistencies. More recently, Stella [2021] extracted English-Spanish and English-German datasets also from Wikipedia biographies that are balanced in gender, occupations, and diversity in nationalities (up to 90). This dataset has been designed to analyze common gender errors such as gender choices in pro-drop, possessives, and gender agreement. The released dataset provides information about the document ID, the source text, the translated text, the perceived gender, the entity name, and the source URL. One limitation of this dataset is that it is only released in two language pairs (English-Spanish and English-German) and contains nine occupations. A structural limitation that is difficult to overcome is that gender is addressed in a binary way since there is little representation of feminine or masculine occupations in Wikipedia. In Renduchintala et al. [2021], authors build SimpleGEN, which is a gender bias test set based on gendered noun phrases. These phrases follow diverse patterns that cover a single, unambiguous, correct answer. Authors specifically calculate the percent of correctly gendered nouns, incorrectly gendered nouns and inconclusive results. Our approach belongs to the category of real-world datasets. The study most similar to ours Costa-jussà et al. [2020] proposes a toolkit to extract multilingual balanced datasets in binary gender (men and women) from Wikipedia biographies. Differently, OCCGEN is customizable in gender categories considering the broad gender spectrum and balanced within occupations. Limitations of these categories are discussed in Appendix H.

**Definitions** In this paper, we define bias as one relevant factor that prevents our systems from being equitable. In this sense, an example of occupation bias when translating from English to French in our translation systems will generate more translations to masculine doctors than to feminine doctors because historically, there are more references to masculine doctors in our data. Even if gender is a spectrum more than a categorical variable D'Ignazio and Klein [2018], in our use cases, we are limiting gender to binary (masculine and feminine), and we are relying on the tagged category of the perceived gender from our sources. We are only considering binary gender in our use cases because of the limited representation of other genders in our data (see Figure 2). The gender categories from this paper are exclusively based on our sources, and we are exempt from any responsibility in this matter since it is out-of-the-scope of this paper to perform any categorization of gender.Interested readers for this issue can refer to wonderful online resources[5]. Moreover, within articles, we also balance in number of sentences. We categorize languages as high-resource or low-resource based on the categorization that has been described by Goyal et al. [2022] where languages with available parallel data less than one million samples are considered low-resource, between one to one hundred millions are considered medium-resource, and greater than one hundred millions are categorized as

---

[3]https://github.com/mt-upc/OccGen_dataset

[4]https://github.com/mt-upc/OccGen_toolkit

[5]`https://www.morgan-klaus.com/gender-guidelines.html`. Our extracted datasets are balanced in gender within occupations in the sense that they have the same number of Wikipedia entities (one entity corresponds to one Wikipedia article) in all genders under consideration for each particular occupation entity. For example, for the case of the *politician* occupation, if limiting to binary categorization of gender (masculine and feminine), we would have $N$ a number of articles for feminine politicians and $N$ for masculine.

high-resource languages. Finally, we provide impact and bias statements and limitations in Appendix H and a data card in Appendix I.

# 3 Proposed Methodology: OCCGEN toolkit

We describe the proposed methodology of the OCCGEN toolkit (summarized in Figure 1).

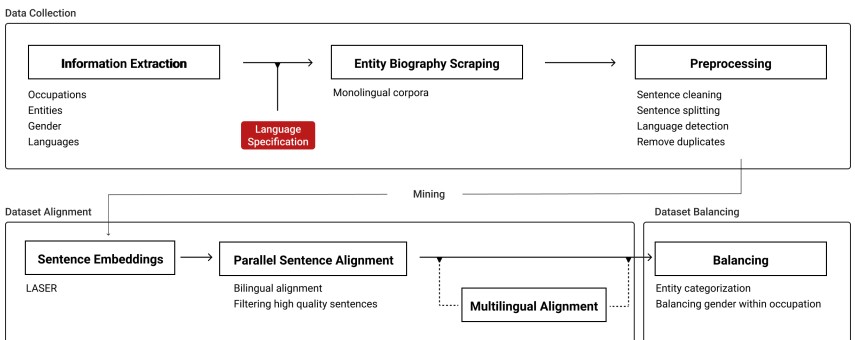

Figure 1: OCCGEN pipeline overview.

## 3.1 Data Collection

On the one hand, our metadata are collected from the Wikidata[6] knowledge base. Wikidata is a project that maintains its data quality by monitoring methods and evaluations to guarantee that it suits users' needs. Briefly, our metadata contains a set of people (*from now on* entities) with their occupation(s), gender, and Wikipedia links in all available languages. On the other hand, the textual data that compose our dataset are extracted from Wikipedia[7], similar to Costa-jussà et al. [2020], Stella [2021]. Textual data consists of the text from the entity's biography for one language from Wikipedia.

**Information Extraction** In this first step, we extract metadata from Wikidata, which relate a set of entities with their working occupations, gender, and biographies from Wikipedia in all available languages. The information extraction procedure is described as follows (see also Appendix A):

1. We extract all the occupations present in Wikidata.
2. For each occupation, we gather the data of every entity that works in the related occupation.
3. For each entity from the previous step, we determine the gender information and related Wikipedia links (corresponding to biographies) in all available languages.

Note that we remove the occupations that do not have related entities and entities that lack gender information. Furthermore, we remove language tags in each entity that do not have a valid ISO language code[8] nor special language code from Wikimedia[9].

**Entity Biography Scraping** At this step, we specify the languages that will be included in our final dataset. The size of the dataset at the end of the pipeline will be heavily influenced by the type and number of selected languages. For instance, high-resource languages are more likely to have more biographies; nevertheless, a multilingual dataset with high-resource languages may significantly reduce the number of sentences compared to a bilingual dataset. As a result, there are implicit trade-offs between high-resource and low-resource languages and between bilingual and multilingual datasets. By specifying a set of ISO language codes to the system, we scrape all the monolingual data from the corresponding Wikipedia biography for entities with a link for all of the given languages.

**Preprocessing** As follows, we describe the steps used to preprocess monolingual data.

---

[6] https://www.wikidata.org
[7] https://www.wikipedia.org
[8] http://www.lingoes.net/en/translator/langcode.htm
[9] https://meta.wikimedia.org/wiki/Special_language_codes

- **Sentence cleaning** Regex expressions are applied to remove the information between brackets and parenthesis, which is mainly related to phonetics, dates, and references.
- **Sentence splitting** Monolingual data are split into sentences; consequently, the sentences are prepared for alignment individually.
- **Language detection** Sentences are fed into a language detection module to exclude those that are not labeled correctly, as Wikipedia pages can mix sentences from several languages, to ensure that all sentences are from the intended language.
- **Remove duplicates** Duplicated sentences are removed to ensure unique sentences for each entity.

### 3.2 Dataset Alignment

Our mining strategy prepares the data so that each entity's data are represented individually. The next steps perform the sentences embeddings of each language independently and compute the candidates sentences between a source and target language on each entity individually. Then, the final data set is obtained from a multilingual alignment.

**Sentence Embeddings** We obtain sentence embeddings of each language through a multilingual sentence encoder based on the architecture Schwenk [2018] in which semantically similar sentences are closer to each other, independent of their language Schwenk et al. [2021]. This allows for a common ground for sentences from different languages. It facilitates the use of the multilingual encoder to extract parallel sentences relying on distance-based metrics to perform the next step parallel sentence alignment.

**Parallel Sentence Alignment** Parallel sentence alignment follows the margin-based criterion introduced in Artetxe and Schwenk [2019] as a metric to execute the nearest neighbor. The margin criterion between two candidate sentences $x$ and $y$ is defined as the ratio between the cosine distance between the two embedded sentences and the average cosine similarity of its nearest neighbors in both directions (equation 1).

$$margin(x,y) = \frac{\cos(x,y)}{\sum\limits_{z \in NN_k(x)} \frac{\cos(x,z)}{2k} + \sum\limits_{z \in NN_k(y)} \frac{\cos(y,z)}{2k}}, \tag{1}$$

where $NN_k(x)$ denotes the $k$ unique nearest neighbors of $x$ in the other language and $NN_k(y)$ denotes the same for $y$. This alignment step allows for getting parallel bilingual candidates which are sorted according to their margin scores, and a threshold is applied to get the desired quality of parallel sentences. This step is performed on each pair of languages independently.

**Multilingual Alignment** In case of multilingual dataset, we consider one language as the target (pivot) to all other languages, and perform a parallel sentence alignment for each pair of languages. For example, for multilingual alignment of English, Arabic, Spanish and Russian, we can consider English as the target language and perform parallel alignment for English-Arabic, English-Spanish and English-Russian (illustrated on Appendix B). Then we obtain the intersection of common sentences between the language pairs depending on the target language (same pivot sentences). The desired quality of these sentences depend on the chosen threshold of the scores of the alignments (illustrated on Appendix B).

### 3.3 Dataset Balancing

We aim to obtain a balanced dataset that will contain the same number of sentences per gender within an occupation.

**Entity Categorization** Entities could have more than one occupation. We categorize entities by the number of occupations they include. Such categorization informs us about the multiplicity of occupations and their corresponding entities in our data. This information enables the choice of categories intended for balancing later. For example, category one represents the entities that have one occupation, category two represents the entities that have two occupations and so on.

**Balancing Gender within Occupation** The output of this step will be a balanced set with respect to numerous occupations. Each occupation will be represented by a similar number of gender entities, and the total numbers of sentences per gender will be the same. There might be an occupation's

name that refers to a single gender, but the data within this occupation will be balanced regarding all the genders (e.g. *actor/actress*). During balancing, we balance each category (i.e., number of occupations) separately and incrementally, for example, balancing category one (i.e., one occupation) followed by category two (i.e., two occupations). Balancing higher categories (i.e., with multiple occupations) means excluding occupations that already exist in previous lower categories. For example, in the case of extracting category two, if we have an entity with occupations of *doctor* and *politician*, this entity is excluded if either *doctor* or *politician* or both were included occupations in category one. This guarantees that the balancing of the new occupations is not conditioned by balancing the occupations of the last category. We continue by computing the number of gender entities for each occupation and the sum of each gender sentence from all the corresponding entities. For each occupation, balancing will be carried out according to the minimum number of entities and sentences. For example, for binary gender (masculine and feminine), if an occupation has four women and five total related sentences and seven men and ten total sentences, then four entities and five sentences are the maximum intended values for each gender in this occupation. Consequently, this step excludes occupations that have one gender representation (feminine or masculine). We prioritize the masculine and feminine entities[10] that have a similar number of sentences with a higher degree of similarity among languages (i.e., this similarity is based on the margin criterion defined in section 3.2). Details are illustrated in Algorithm 1 for the specific case of using binary gender as we do later in our use cases.

---

**Algorithm 1:** *Balancing gender within occupations.*

**Input** :$U_{dic}$ // An unbalanced dictionary containing the information about occupations, entities in each gender, and aligned sentences with their alignment score.

**Output**:$B_{dic}$ // A balanced dictionary where each occupation has the same number of sentences in balanced masculine and feminine entities.

1   $Occs$;// A list of occupations in $U_{dic}$.
2   $Em_i; Ef_i$;// A list of masculine and feminine entities with $i$th occupation, respectively.
3   $Sm_i; Sf_i$;// Number of sentences in $Em_i$ and $Ef_i$, respectively.
4   $B_{dic} = \{\}$;// Initialize the empty dictionary to store the balanced information.
5   **for** $i \leftarrow 0$ **to** $len(Occs)$ **do**
6      **if** $len(Em_i)==len(Ef_i)$ **then**
7          Balance the entities from $Em_i$ and $Ef_i$ such that $Sm_i$ is equal to $Sf_i$ and update $B_{dic}$;
8      **else**
9          $Emin = \min(len(Em_i), len(Ef_i))$;
10         **if** $Emin==len(Em_i)$ **then**
11             Select only $Emin$ feminine entities with high-quality sentences from $Ef_i$;
12             Balance the entities from $Em_i$ and $Ef_i$ such that $Sm_i$ is equal to $Sf_i$ and update $B_{dic}$;
13         **else**
14             Select only $Emin$ masculine entities with high-quality sentences from $Em_i$;
15             Balance the entities from $Em_i$ and $Ef_i$ such that $Sm_i$ is equal to $Sf_i$ and update $B_{dic}$;
16         **end if**
17      **end if**
18 **end for**
19 **return** $B_{dic}$

---

# 4   Use-case Study

In this section, we report the experimental details of our methodology by including details on two use cases (high- and low-resource languages) limited to balancing in binary gender (masculine and feminine).

## 4.1   High-resource languages

The top-7 languages with the largest number of entities at Wikipedia are English, German, French, Spanish, Russian, Italian, and Arabic. Among these top languages, there are four linguistic families, Germanic, Latin, Slavic, and Semitic, and we choose one language representing each family. We extract multiparallel data among the high-resource languages that cover different linguistic families, including Semitic (Arabic, ar), Germanic (English, en), Slavic (Russian, ru), and Latin (Spanish, es). The motivation of this use case is to have a balanced dataset in languages that are well-studied in the community. Nonetheless, this dataset can also be used in conjunction with other existing benchmarks

---

[10]Note that we use the terms feminine/masculine and women/man coherently with linguistic praxis instead of the Wikipedia category female/male

that may contain occupational stereotypes or unbalances in gender. Hereinafter, we alternatively refer to this use case either as the high-resource or en-es-ru-ar use case. The latter mentions specifically the covered languages.

## 4.2 Low-resource languages

We extract parallel data for the bilingual case of Swahili (sw) and English (en). We choose Swahili to represent low-resource African languages because it has moderate coverage in Wikipedia and is supported by the multilingual sentence encoder we use in the alignment process. The motivation of this use case is to have a balanced benchmark in both occupations and gender for low-resource languages and increase the representation of African languages in the NLP community. Hereinafter, we alternatively refer to this use case as either the low-resource or en-sw use case. The latter explicitly mentions the covered languages.

## 4.3 Implementation details

We extract data from Wikidata using a Python SPARQL wrapper. For entity biography scraping, we implement an algorithm that works with Beautiful Soup[11], whose purpose is pulling data from HTML content. After that, the following preprocessing techniques are implemented to improve the outcome of our collection process:

- We use regex expressions to clean the collected textual data.
- We use the nltk[12] sentence tokenization package[13] to split sentences across all languages except Arabic, which uses a sentence splitter wrapper[14] for CoreNLP[15].
- We apply language detection to sentences using Compact Language Detector 3[16], which can identify up to 108 languages, to remove sentences that are not labeled with the appropriate language.
- We also remove sentences repeated within an article.

We prepare the text for each entity individually for mining. Then, to execute parallel sentence alignment, we utilize LASER Schwenk and Douze [2017], which provides multilingual sentence embeddings. After embedding the sentences, the aligned parallel sentences in a language pair are computed using the distance in the embedding space. Candidates are sorted according to the order of the similarity between sentences. When aligning multiple languages, English is the pivot language. Finally, note that both datasets, after being automatically extracted, are postedited to be used as evaluation benchmarks. Details on the human postedition are on the Appendix C.

## 5 Experimental Results

In this section we report the experimental results of the use cases that we proposed.

## 5.1 Data Statistics

**Entities per language.** Figure 2 shows the number of entities for each language and gender. We see the difference between high-resource languages and low-resource languages. English is the language with the highest number of entities, and Swahili is the language with the lowest. There is a large difference between gendered representations. We observe very few entities that are not man or woman. The figure shows that all languages have three times more masculine representations than feminine.

**Number of entities, occupations and sentences through the pipeline.** Table 1 shows the number of entities, occupations, and sentences at the different stages of our pipeline (Figure 1): entity biography scraping, preprocessing, alignment (multilingual or bilingual), and balancing. These statistics show how the number of entities and occupations is reduced at each step for our user cases. They show that alignment and balancing steps have a great impact in reducing the number of occupations and entities,

---

[11] https://www.crummy.com/software/BeautifulSoup/bs4/doc/

[12] https://www.nltk.org

[13] https://github.com/Mottl/ru_punkt

[14] https://github.com/chaojiang06/CoreNLP_sentence_splitter

[15] https://stanfordnlp.github.io/CoreNLP/index.html

[16] https://github.com/google/cld3

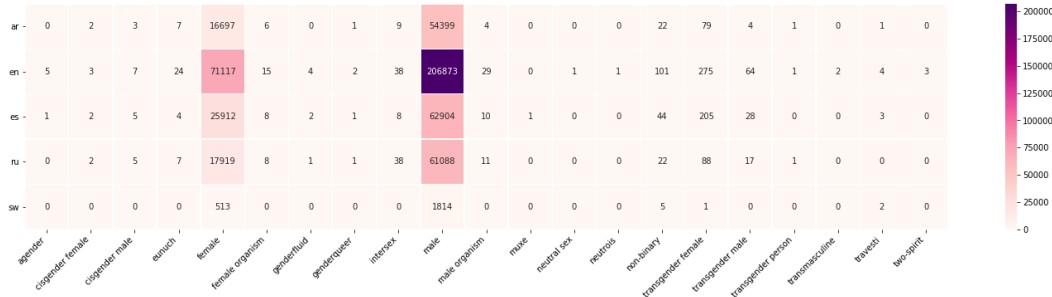

Figure 2: Distribution of entities' gender across languages.

illustrating the reason for losing entities and corresponding sentences when choosing more languages to align and balance. The numbers of sentences can only be provided from the alignment step onward, since sentences per language can only be noted individually before this step. As predicted in section 3.3, among the balanced occupations in the high-resource use case, we found occupations' names characterizing only the masculine gender, such as *pornographic actor* or *monarch*. Appendix E reports details on the statistics of entity categorization explained in section 3.3.

|  |  | Entity biography scraping | Preprocessing | Alignment[*] | Balancing |
|---|---|---|---|---|---|
| **en-es-ru-ar** | entities | 15421 | 14635 | 2436 | 286 |
|  | occupations | 644 | 256 | 203 | 59 |
|  | sentences | - | - | 6732 | 524 |
| **en-sw** | entities | 1647 | 1371 | 1053 | 277 |
|  | occupations | 281 | 281 | 73 | 38 |
|  | sentences | - | - | 5426 | 730 |

Table 1: Evolution of the number of entities and occupations through the pipeline. [*]Multilingual alignment, en-es-ru-ar and bilingual alignment, en-sw.

## 5.2 Machine Translation

**System description and implementation** To evaluate our dataset, we used the downstream MT task. We used three multilingual models that include the languages from our use cases: M2M_100 Fan et al. [2020], mBART50_m2m Tang et al. [2020] and Opus-MT Tiedemann and Thottingal [2020]. These systems are transformer-based models Vaswani et al. [2017], and they use SentencePiece-based segmentation Kudo and Richardson [2018]. M2M_100, supports translation between any direction for 100+ languages, includes many-to-many supervised training covering thousands of language directions. mBART50_m2m supports translation between any direction for 50+ languages and it has been trained with supervised translation from and to English. Opus-MT supports 1200+ translation directions for 150+ languages.We used the default implementation from EasyNMT[17].

**Results** Figure 3 reports the results in terms of BLEU for all (top) and the two feminine/masculine subcorpora from our high-resource use case benchmark. See Appendix F for all results in all translation directions. For English and Arabic (both directions) and translating to Russian, the performance is better (or similar) in the masculine set for all models. In the case of translating from Russian, the performance of the feminine subdataset is better than that of the masculine subdataset in all models except for the mBART50_m2m model. When translating to Spanish, this improvement only holds for the M2M_100 model.

For the low-resource languages use case, the results are reported in Figure 4. The M2M_100 model performs the best, and similar to high-resource languages, mBART50_m2m performs the least. We could not obtain the results for Swahili-to-English with Opus-MT models since it does not support this direction.

---

[17]https://github.com/UKPLab/EasyNMT

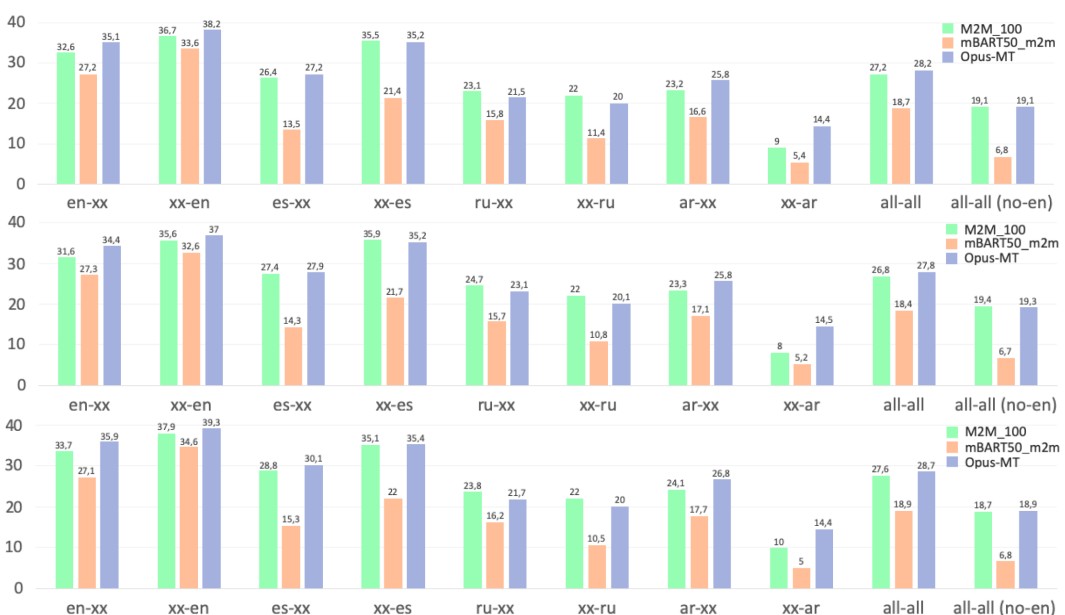

Figure 3: High-resource language results. Average BLEU for M2M_100, mBART50_m2m and Opus-MT models. (Top) All, (Mid) Woman (Bottom) Man.

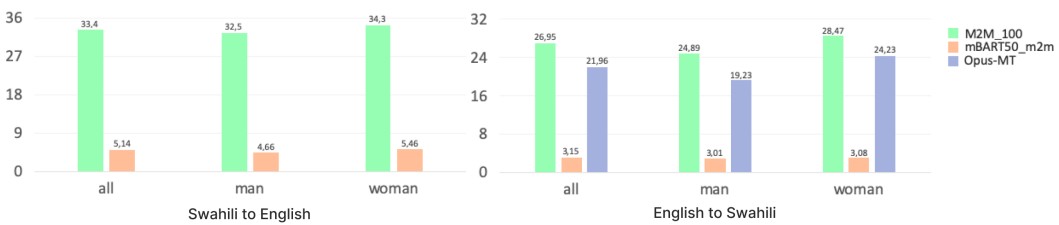

Figure 4: Low-resource language results. BLEU scores for the Swahili to English and English to Swahili direction with M2M_100, mBART50_m2m and Opus-MT models. Note that for Swahili to English, results for Opus-MT model were not obtained because it does not support this direction.

**Discussion** There is an intriguing pattern that shows that masculine English translations improve feminine set in the all-all performance but feminine translation improve masculine set in the case of all-all (no-en). Furthermore, if we analyze performance with English either on the source or the target side (en-xx or xx-en) on both genders (Figure 3 bottom mid), we found that the difference between the performances of masculine and feminine translations using English on either side is the greatest for gender comparisons in any other high-resource language pair, which explains the change in translation performance with and without English. This reveals that the performance in English may be skewed towards masculine translations in any direction.

When looking at the translation direction (e.g., language A to all or all to language A), we observe English and Spanish to exhibit differed behaviors than Russian and Arabic. The former languages exhibit a higher BLEU performance when the languages are on the target side rather than on the source side. For the same languages, on the target side, the performance on the feminine set tends to be lower than the masculine set. We hypothesize that English and Spanish have a solid language generation. This strong language generation might be, in part, due to the fact that they are written in Latin script, which is shared among many high-resource languages (i.e., Italian, French, and Portuguese). Having a sound language generation may help to achieve higher performance when on the target side. However, it may also contribute to paying less attention to the source sentence and more attention to the previous words in the target sentence, which may explain why the performance in the feminine set did not improve. Less attention given to the source sentence and more attention given to the previously generated words in the target can overgeneralize to the most frequent gender, which tends to be masculine. This suggests that when translating to English, even if source languages (Spanish,

Russian, and Arabic) have high morphological information, the performance on the masculine set is higher than that on the feminine set. This could be explored and confirmed with interpretability measures Ferrando and Costa-jussà [2021], Ferrando et al.. For the latter language set, Russian and Arabic, the performance of the translation direction does not vary as it does in the previous case. This may be because even if they are high-resource languages, they have Cyrillic and Arabic scripts, respectively, which are not shared among other high-resource languages. This may explain the poorer language generation.

Regarding the low-resource use case in Figure 4, the performance in the feminine set is better than that in the masculine set for all models and directions. Both Swahili and English are low-inflected languages in gender, which means we are analysing a language pair which is less prone to having gender bias. The results obtained for Swahili-to-English are better than those obtained for English-to-Swahili, probably due to the language generation for English being better than that for Swahili. The large difference between the performances of the high-resource and low-resource use cases might be caused by the few Swahili data included in the training corpora used to pretrain the used models. Furthermore, the fact that the Swahili-to-English results obtained with the Opus-MT model were not obtained using EasyNMT even though Swahili was claimed to be supported also reveals the challenges of working on low-resource languages, thus the NLP community should increase their representation.

**Human Evaluation**    For human evaluation of gender accuracy, we marked the critical gender word of the sentence in the English part of the high-resource dataset. For example, in the sentence *Sopita Tanasan is a Thai weighlifter.*, the word *weighlifter* was marked. For all language pairs in this set, native annotators marked if the gender of the marked words was correctly translated. In the previous example, when translating to Spanish, the output was *Sopita Tanasan es un levantador* de pesas tailandés.* Since the gender of the marked word is wrong (masculine, *levantador*, instead of feminine, *levantadora*), the sentence is annotated as incorrectly translated. See more details on the annotation in Appendix section D.

We computed gender accuracy as the number of correctly translated gender of marked words (each representing its sentence) divided by the total number of marked words. We conducted the human evaluation for the translation directions of English-to-Arabic, English-to-Russian and English-to-Spanish. Overall, the results agree with the automatic evaluation, where the feminine accuracy is lower than the masculine accuracy for these specific translation directions. The highest feminine accuracy is found in the Russian translations, where the feminine accuracy reaches 0.68 while the masculine accuracy is 0.81. On the other hand, the lowest feminine accuracy occurs in Arabic translations, where the feminine accuracy is 0.41. However, the masculine accuracy is much higher, reaching 0.78. The accuracy of Spanish translations difference between man and woman is the least, where the feminine accuracy is 0.62, and the masculine accuracy is 0.71.

# 6    Conclusions

This paper proposes the OCCGEN toolkit to generate monolingual, bilingual, and multilingual balanced datasets in gender within occupations. This freely available toolkit is customizable in languages and gender. Our toolkit simplifies the extraction of parallel data and it is already been used by the community Zhou [2022], Liu [2022]. We present a high and low-resource benchmark. The former includes a multilingual English, Spanish, Russian, and Arabic dataset. The latter includes a bilingual dataset on English-Swahili translations. Both of them are balanced for the particular case of binary gender (masculine/feminine). Note that with our benchmarks, we have postedited the output of the OCCGEN toolkit to provide an accurate evaluation set. However, we can also use our toolkit to extract training data which is not necessary to postedit. Appendix G provides this kind of data for the English-Arabic, English-Spanish and English-Russian pairs. Differently from the provided benchmark, we have not balanced this data in order to have as many data as possible, covering all genders available in the Wikipedia for the extracted entities. We suggest that this data can be balanced by means of artificial techniques such as oversampling, counterfactual techniques or other synthetic techniques in future. Once balanced, this data can further be used to fine-tune models in order to mitigate gender biases as conducted in previous studies Saunders and Byrne [2020], Costa-jussà and de Jorge [2020]. Our toolkit, benchmarks and training data are released in Github as referenced in section 1.

We report experiments using our benchmark datasets to evaluate MT models. We provide an accurate analysis of performance behavior for the particular case of binary gender. We conclude that feminine translations tend to be worse for high-resource languages with a high-quality language generation model. We hypothesize that, in these cases, the model gives less attention to the source words than the target prefix, and using the target context may overgeneralize to most frequent patterns (which tend to be masculine patterns) rather than producing an accurate translation. This is confirmed by the human evaluation that computes gender accuracy on marked words (for English to Arabic, Spanish, Russian), which shows a higher accuracy for the masculine cases than feminine cases.

Our balanced datasets are not strictly comparable across genders since each gendered subset has different vocabularies. Therefore, a reasonable further step is to automatically generate counterfactual data Qian et al. [2022]. This would modify the masculine subdataset into a feminine subdataset and vice versa. Ultimately, with this data augmentation, our balanced sets would allow for the analysis of gender performance with standard evaluation methods and without requiring new ones.

## Acknowledgments and Disclosure of Funding

Authors would like to specially thank annotators for tasks described in Appendix C and D: Carlos Escolano, Gerard Gallego, Ksenia Kharitonova, Gerard Sant, Elena Zotova, Marianne Raouf, Merna Onsy, Engy Basta, Delphina Ndekupatia Severin, Perez Ogayo, and Lawrence Ilagoye Gwajekale. Authors appreciate the insightful comments received by the anonymous reviewers. Authors are extremely thankful to Pascale Fung and Kendra Albert for their valuable ethics discussion; to Christophe Ropers and to Eric Smith for their feedback; and to Delice Musabyimana Agahozo for helping in contacting some Swahili annotators. The work is partially supported by Research Grants (AGAUR) through the FI PhD Scholarship, Universitat Politècnica de Catalunya with the collaboration of Banco de Santander.

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
