# A    Information Extraction

Figure 5 shows an schema explaining the extraction of the entities. Each step is depicted in a triplet format: ⟨subject,predicate,object⟩. *Blue* (italics) information is the information extracted at each step. For each step outlined with a dotted rectangle (−−), the information extracted is the subject; otherwise, the information extracted is the object.

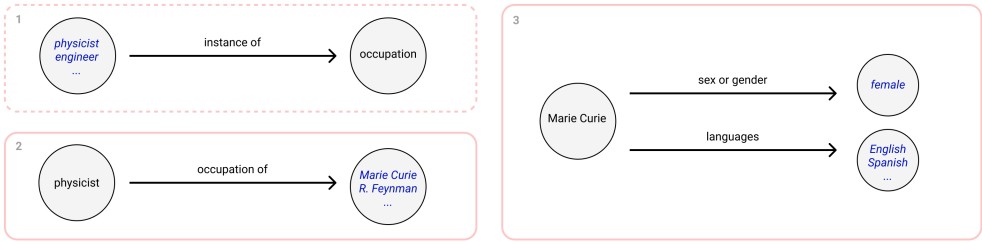

Figure 5: Extraction schema.

# B    Multilingual Alignment

Figure 6 show an example of multilingual alignment for the languages considered in the high-resource use case: English, Arabic, Spanish and Russian. In this case, we have English as the pivot language in all parallel alignments, so intersection is computed with common English sentences (green) to obtain the aligned sentences in all four languages.

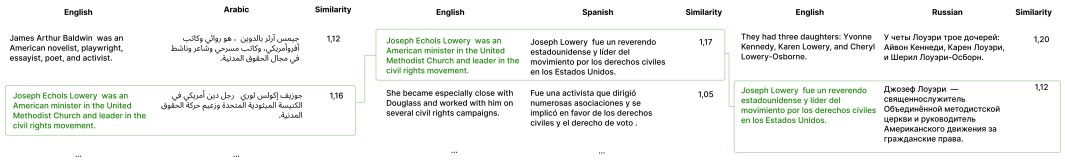

Figure 6: Multilingual alignment.

For each multilingual alignment, the LASER similarity has to be above a desired threshold. Figure 7 shows an example of this quality filtering. In this example, we have set a threshold of 1,1 points. Thus, if one parallel alignment is below this number, then, we do not consider the multilingual alignment in further steps as in the right example (red background).

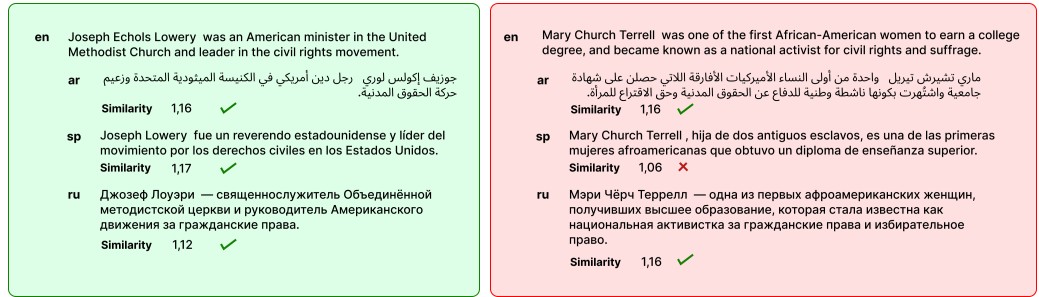

Figure 7: Quality filtering.

# C    Postediting

Given the high-resource and low-resource datasets, we postedit them to have curated datasets that can be used for evaluation in machine translation. We use English as the anchor language and distribute

sentences in a spreadsheet (see Figure 8) for native annotators in non-English languages in which English is the second language.

| | A | B | C | D | E |
|---|---|---|---|---|---|
| 1 | **English** | **Spanish** | **Check** | **Post-Edit** | **Perceived Gender** |
| 2 | Sopita Tanasan  is a Thai weighlifter. | Sopita Tanasan  es una levantadora de pesas tailandesa. | M | No | female |
| 3 | Abeer Abdelrahman Khalil Mahmoud  is an Egyptian weightlifter. | Abeer Abdelrahman Khalil Mahmoud  es una halterófila egipcia. | M | No | female |
| 4 | Romela Aleksandër Begaj is an Albanian weightlifter. | Romela Begaj  es una una halterófila albanesa. | M | Yes | female |
| 5 | Cao Lei  is a Chinese weightlifter. | Cao Lei es una halterófila china. | M | Yes | female |
| 6 | Jennifer Lombardo  is an Italian weightlifter who won two gold medals at the 2018 Mediterranean Games. | Jennifer Lombardo  es una halterófila italiana que ganó dos medallas de oro en los Juegos Mediterráneos de 2018. | M | Yes | female |
| 7 | Margaryan won a bronze medal at the 2010 Summer Youth Olympics. | Margaryan ganó una medalla de bronze en los Juegos Olímpicos Juveniles de Verano de 2010. | M | Yes | male |
| 8 | Long Qingquan  is a Chinese weightlifter. | Long Qingquan es un halterófilo chino. | M | Yes | male |
| 9 | Kianoush Rostami  is an Iranian Kurdish Olympian weightlifter. | Kianoush Rostami  es un levantador de pesas iraní. | M | No | male |
| 10 | Nurudinov also won a gold medal at the 2016 Olympics, setting a new Olympic record in the clean and jerk at 237 kg. | Nurudinov también ganó una medalla de horo en los Olímpicos de 2016, marcando un nuevo record olímpico en cargada y envión de 237 quilos. | M | Yes | male |

Figure 8: Spreadsheet for annotators. Complete example for the Spanish language.

Each language set of sentences was split into 2 to 4 subsets addressed by different annotators. The annotation guidelines are as follows:

*Given a sentence in English (first column (A)) and Arabic/Russian/Spanish/Swahili (second column (B)), do the minimum number of edits (in the same column (B)) to the Arabic/Russian/Spanish/Swahili sentence to match the English sentence. If the Arabic/Russian/Spanish/Swahili sentence contains more information than the English sentence, then remove it. If the English sentence contains more information, add this information translated into Arabic/Russian/Spanish/Swahili. Mark each sentence that is edited (add M to the third column (C)). If you do not know how to postedit without changing the meaning, mark "NM" in the third column (C). Mark if postediting was necessary (yes) or not (no) in the fourth column (D). Pay special attention to gender; if it is ambiguous, please check the entity-perceived gender from the fifth column (E).*

After annotating the entire dataset in each language, there was an additional annotator for each language who reviewed the entire set. Annotators were volunteers, and they are acknowledged at the end of this work. For the African language, annotators were contacted through the Masakhane community[18].

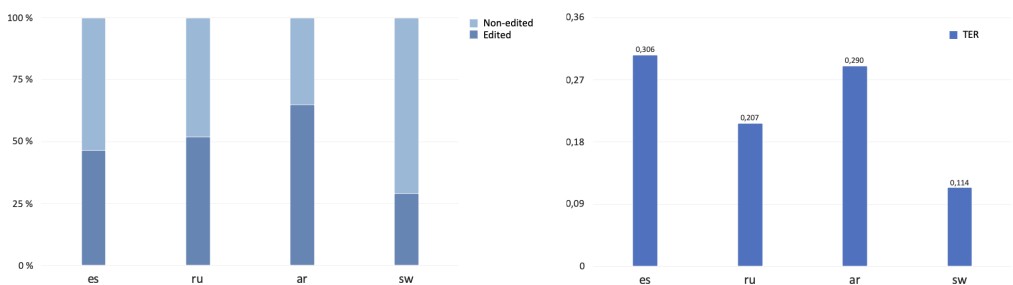

Figure 9: Percentage of postedited sentences per language (left). translation edit rate (TER) per language (right)

Figure 9 (left) shows the number of postedited sentences from high- and low-resource languages. Note that sets in Spanish, Russian and Arabic are comparable, but sets in Swahili are different. For high-resource languages, the proportion of sentences that need to be postedited is approximately 50%. For low-resource languages, this proportion is smaller, approximately 30%. Figure 9 (right) shows the translation edit rate (TER) results computed with Huggingface's whitespace tokenization. Results are coherent with the previously postedited sentences. The low TER in the Swahili case is interesting. This TER gives us an idea of the error that our toolkit can introduce when extracted data are not postedited. The TER values, which are not greater than to 30% in any case, show that the amount of postediting is moderately low. Moreover, assuming this error, our toolkit can be considered for training purposes without requiring postediting as we release and explain in Appendix G.

---

[18]https://www.masakhane.io/

## D    Human Evaluation

To go beyond BLEU evaluation, we performed a human evaluation to quantify the accuracy of the translated gender. As our data is extracted from Wikipedia biographies, sentences usually have an occupation, pronoun or a verb referring to the main character. We marked one main gendered part in each sentence, as following:

1. If the sentence has an occupation, we mark the occupation.
2. If the sentence does not have an occupation, we mark the primary pronoun of the sentence. If the main pronoun occurs with a verb, we mark the pronoun and the verb together (e.g., *she won*). In case of no pronoun but a verb, we mark the verb accompanying the main character. In other cases of pronouns, we mark the pronoun only (e.g., *he, her, him*).

This annotation was done in the English part of the high-resource dataset and it is released together with our data.

The annotator is provided with pairs of sentences which include one English sentence and its corresponding translated sentence (either in Arabic, Spanish or Russian). Each English sentence has one or several bold words. Then the human evaluators were provided with the following guidelines:

*Mark each pair as 'Correct', 'Incorrect' or 'N/A'. The pair is correct if the gender of the correspondence of the bold words in the translated sentence matches the perceived gender and incorrect if not. The pair is n/a if the gender is not specified both in source and target. Several edge cases to consider: if the gender is not explicit in the source but is explicit in the target, it is considered correct in case it matches the perceived gender, otherwise, considered incorrect. In another case, if the gender is specified in the source but not in the target (because it is not necessary to specify it), it is considered correct. In the case of pronouns, usually, the pronouns refer to the perceived gender. However, in a few cases, the pronoun in the sentence refers to another character; then, this pronoun has to be checked if correctly translated or not according to the sentence.*

Evaluation was performed for English-to-Arabic, Russian and Spanish. Each pair was evaluated by one different human annotator. Annotators were native in the target language and proficiency in English. Results of this human evaluation are commented in section 5.2.

## E    Entity Categorization

Figure 10 shows the number of entities with different amounts of occupations regarding our use-cases: (left) English-Spanish-Russian-Arabic (en-es-ru-ar) (high-resource), and (right) English-Swahili (en-sw) (low-resource).

## F    Heatmap Results

Figure 11 shows the heatmap of BLEU results for different language pairs. In general, we see the best performance for the Opus-MT model, with few exceptions (English-Russian, Spanish-Russian, and Spanish-to-English) on which M2M_100 is better. mBART50_m2m has the lowest performance, especially in directions that do not involve English, which makes sense because it is unsupervised. The best results are obtained when translating to English, and the worst results are obtained when translating to Arabic.

## G    Training data

We are releasing raw data for the English-Arabic, English-Russian and English-Spanish pairs. By raw data, we mean that we use OCCGEN toolkit and we do not postedit the output. This new data set include all the gender categories with non-zero entities from the Wikipedia sources in these languages, as shown in Figure 12. However, this set of additional data now does not have the same number of entities in all gender categories. As mentioned in the conclusions 6, this data, in the future, can be balanced by means of artificial techniques. Then, it can be used to mitigate gender biases by using it for fine-tuning models trained on unbalanced data Saunders and Byrne [2020], Costa-jussà and de Jorge [2020].

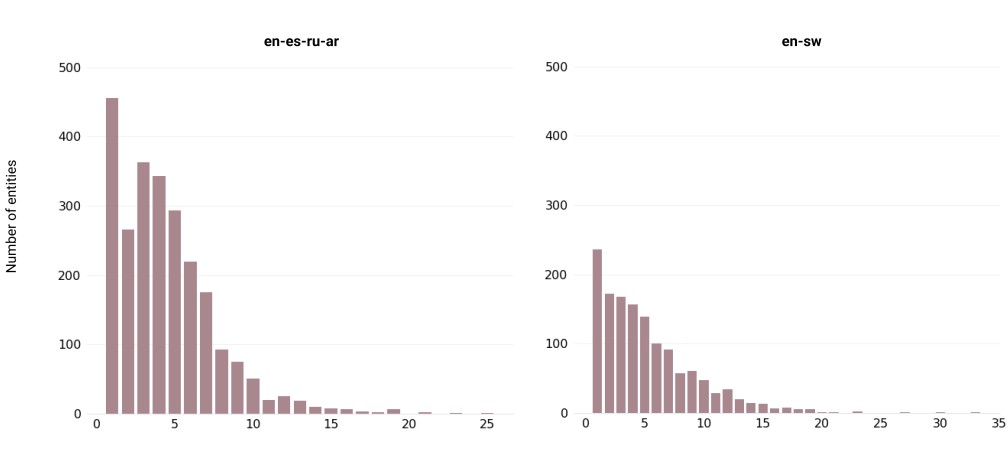

Figure 10: Number of entities with different amounts of occupations regarding our use-cases.

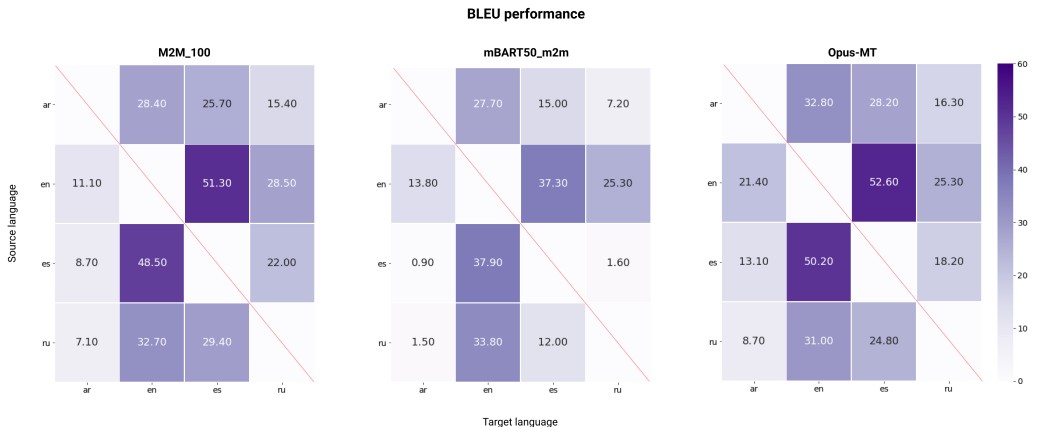

Figure 11: High-resource language results. Heatmap for BLEU scores between languages with M2M_100, mBART50_m2m and Opus-MT models.

As follows we report the exact details of the data extraction. We extracted training data for English-Arabic, English-Russian and English-Spanish language pairs. We carried out the pipeline for each pair of language individually. We applied the same pipeline of data collection in section 3.1. For any language pair, the entities extracted are those with non-zero values for any available gender in both languages of the pair (see Figure 12), i.e. to enable alignment between the language pair. Then we applied dataset alignment as described in section 3.2. We specified the accuracy of the alignment to be greater than threshold one. We did not apply dataset balancing to keep a reasonable amount of data while keeping beyond masculine and feminine gender categories. After this procedure, we got 122,916 sentences for English-Arabic, out of which 1,378 were extracted for genders which were neither masculine nor feminine ones. For English-Russian, we got 433,537 sentences, out of which 3,889 were extracted for genders which were neither masculine nor feminine. we got 1,048,530 sentences for English-Spanish, out of which 5,101 were extracted for genders which were neither masculine nor feminine ones.

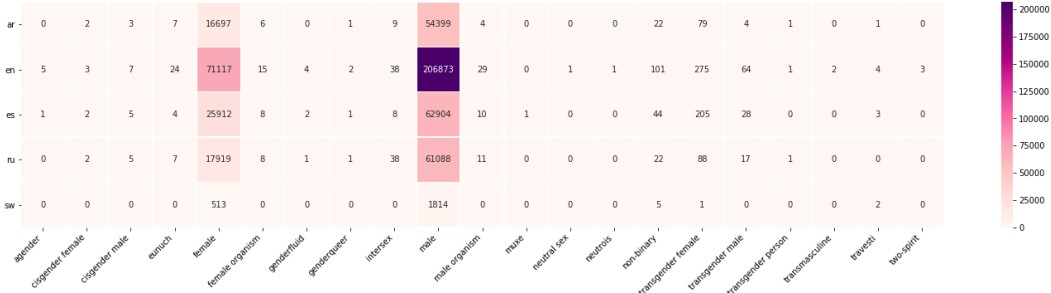

Figure 12: Distribution of entities' gender across languages in the training data.

# H   Impact, Bias Statements and Limitations

**Impact statement**   Our work focuses on raising awareness of the imbalances in the datasets commonly used in natural language processing. At the same time, we propose addressing these imbalances in gender within occupations by providing a free multilingual toolkit that can extract data from Wikipedia. Moreover, we release human-annotated balanced benchmarks for several high and low-resource languages (for the particular case of masculine and feminine genders). The impact of using balanced datasets will help in building applications that have similar performance for different social groups Saunders and Byrne [2020], Costa-jussà and de Jorge [2020].

**Bias statement**   Devinney et al. [2022], Hardmeier et al. [2021]. The OccGen toolkit extracts balanced datasets in gender within occupations. Our methodology extracts data from Wikipedia Biographies, therefore, gender balancing is limited to the gender categories that are represented in this data source. Moreover, an additional limitation is the low representation of several gender categories. Therefore, in practice, if we prioritize to extract a large amount of data, as we do in our use cases, we have to limit the gender representation to binary gender masculine and feminine. However, if this is not the case, our methodology can extract lower amounts of balanced data beyond binary gender. Our released data is limited to binary gender. However, our data has the advantage that it can make progress in more accurately representing binary cases (feminine and masculine). Our use case is specially useful to face bias errors in languages with grammatical gender (i.e. Arabic, Spanish, French, Russian) but also in languages with natural gender (i.e. English). More details on classification of how languages treat gender Basta [2022]. Along our paper, for binary gender we use the linguistic praxis of man/woman and masculine/feminine, instead of using the categories from Wikipedia (male/female). There is the exception of the cases which we consider is more appropriate to be faithful to the exact labels of Wikipedia which are Figure 2 and 12 and in the released data itself (shown in Figure 9 and described in the data card in Appendix I). Our source data comes only from the Wikipedia domain but it can be used in any application of natural language processing. This means that both the advantages and limitations of our extracted data can impact any of those applications.

**Limitations**   One of the limitations of our tool is that the gender categories from this paper are exclusively based on the Wikipedia taxonomy which may have its limitations compared to other existing resources as discussed in section 2. Another big limitations is that the size of the extracted data is limited to the amount of available data in Wikipedia biographies. More explicitly, the amount of data is lower bounded by the amount of the minority gender data. This means that a parallel dataset in four languages, and balanced in all genders with non-zero entities from Wikipedia sources, would be very small (see Figure 2). This is a reflection of societal bias in the natural data. Therefore, our use cases have to be limited to binary gender (masculine and feminine) in order to include a larger amount of sentences. However, even with this limitation, our proposed multilingual dataset with binary gender (feminine/masculine) balance is an important first step on the path of progress away from a male-dominated gender bias.

# I    Data card

**Data Card Details**

- Title: OCCGEN_HRLR_dataset
- Summary: The OCCGEN_HRLR dataset has been designed to be gender-balanced in occupations, contain document-level information and allow for an evaluation free of stereotypes in occupations for 4 high-resourced multiparallel languages (Arabic, English, Russian, and Spanish) and 2 low-resourced language pairs with English (Swahili). The set has been designed to evaluate on a balanced dataset
- Publisher: Universitat Politècnica de Catalunya
- Funding: European Research Council (ERC) under the European Union's Horizon 2020 Research and Innovation Programme (grant agreement no. 947657).
- Authors: Marta R. Costa-jussà, Christine Basta, Oriol Domingo, André Niyongabo Rubungo

**Data General Motivation and Access**

- Motivation: Evaluation balanced in gender and occupations
- Application: Machine Translation and other Natural Language Processing Applications.
- Access: Available at `https://github.com/mt-upc/OccGen_dataset`

**Primary data types and preprocessing tools**

- Data type: Nonsensitive public data about people
- Source/target text: Wikipedia biographies
- Tools: Data aligned with LASER Schwenk and Douze [2017]

**Dataset snapshot**

|  | (Ar, En, Ru, Es) | (Sw, En) |
|---|---|---|
| Total instances | 526 | 712 |
| Male/female entities | 286 | 216 |
| Occupations | 59 | 33 |
| Primary data modality | Textual data | |

**Example: HR**

- English; Sopita Tanasan is a Thai weightlifter.
- Arabic: سوبيتا تاناسان رافعة أُثقال من تايلند.
- Russian: Сопита Танасан — тайская тяжелоатлетка.
- Spanish: Sopita Tanasan es una levantadora de pesas tailandesa.
- WikiData_id: Q16227761
- Entity_name: Sopita Tanasan
- Occupation_name: weightlifter
- Perceived_gender: female
- url: `https://en.wikipedia.org/wiki/Sopita_Tanasan`

**Example: LR**

- English: Khadija Gbla is a feminist and human rights activist from Sierra Leone.
- Swahili: Khadija Gbla ni mwanaharakati wa wanawake na haki za binadamu kutoka Sierra Leone.
- WikiData_id: Q61283864
- Entity_name: Khadija Gbla
- Occupation_name: human rights activist
- Perceived_gender: female
- url: `https://en.wikipedia.org/wiki/Khadija_Gbla`

**License and status**

- License: CC-BY-SA 3.0
- Status: Limited maintenance, possible extension to more languages.
- Version: 1.0
- First Edition: Last updated/First Release first half 2022

**Data collection and selection methods**

- Collection: Scraped from Wikipedia.
- Selection: Language and perceived gender tag available. Four languages with a higher number of entities and different linguistic family. An African language with English and a higher number of entities.
- Excluded: No language or perceived gender tag available. Rest of the languages.

**Sampling**

- Methods: Sentence alignment with LASER.
- Automatic criteria: LASER threshold, preprocessing, balancing in gender and occupations.

**Human Attributes Statistics**

- Perceived Gender
- Occupations

**Labelling methods**

- Scraped labels: Perceived Gender, Language, Occupation
- Automatic labels: Sentence alignment with LASER
- Human Annotation Guidelines: Given a sentence in English (first column (A)) and Arabic/Russian/Spanish (second column (B)), do the minimum number of edits (in the same column (B)) to the Arabic/Russian/Spanish sentence to match the English sentence. If the Arabic/Russian/Spanish sentence contains more information, remove it. If the English sentence contains more information, add this information translated into Arabic/Russian/Spanish. Mark each sentence that is edited (add M to the third column (C). If you do not know how to post-edit without changing the meaning, mark "NM" in the third column (C). Mark if postediting was necessary (yes) or not (no) in the fourth column (D). Pay special attention to gender; if it is ambiguous, please check the entity gender from the fifth column (E)