# OpenReview forum: "OccGen: Selection of Real-world Multilingual Parallel Data Balanced in Gender within Occupations"
_NeurIPS.cc/2022/Track/Datasets_and_Benchmarks — NeurIPS 2022 Datasets and Benchmarks _

### Official Review · Reviewer_Lzu8 · 2022-07-20
**The paper mainly presents a toolkit and two case studies while datasets and benchmark tasks, evaluation metrics are not clearly presented and discussed.**

**Rating:** 4
**Confidence:** 5
**Correctness:** 1. The claim on mitigation with the c…

**Strengths:**

1. Gender bias is an important issue in machine translation. The paper presents a toolkit to collect datasets for evaluating this issue.

2. The methodology of the toolkit together with implementation details in high/low-resource languages are presented.

3. Two case studies with experiment results are reported.



**Weaknesses:**

1. The sizes of the presented datasets for high/low-resource languages are small, which makes the sufficiency of using the collected data to evaluate gender bias in machine translation skeptical. It would be much better if the paper could discuss the size issue, e.g., which factor has a great impact of the final size of the collected dataset, whether the toolkit can be used to collect sufficient instances for languages that do not have many entities in Wikipedia.

2. Although details of the toolkit are presented, OccGen seems to be quite sophisticated and manual postedition is required to build the final benchmark dataset, which could make it difficult to adapt the toolkit to other languages.

3. It is not clear how the work presented in the paper is suitable to this track which prioritizes datasets and benchmarks while the paper seems to focus on a toolkit. Intrinsic analysis on the collected datasets and clear definition of benchmark tasks on the collected datasets seem to be missing.

4. The paper states that it attempts to address both evaluation and mitigation issues. However, it is not clear how gender bias is mitigated in machine translation with a small collected balanced dataset.

5. The analysis presented in experiments seems to be language-dependent. It is hence not clear whether the conclusion from this analysis could be extended to other languages.

6. The analysis on the more/less attention to the source sentence and words in the target sentence seems to be short of empirical evidence. Visualization or deep look into NMT models on gender generation could be used as the support to this conclusion.



**Additional Feedback:**

Writing of the paper could be improved, e.g., line 24 & 25.

**Clarity:**

The paper is not clearly written on some key issues. Please refer to comments before.


**Documentation:**

The paper presents a data card for dataset documentation.

**Ethics:**

N.A.

**Relation To Prior Work:**

Although the paper theoretically compare with previous work in the section of related work, it could be better if detailed comparisons to previous datasets on gender bias, e.g., size of dataset, aspect involved in the built dataset, are presented.

**Summary And Contributions:**

The paper presents a toolkit, OccGen, to collect a multilingual parallel and balanced dataset that contain a similar number of sentences per gender within each occupation, from Wikipedia. The pipeline-style toolkit consists of data collection (including information extraction on entities, occupations, gender and language information, biography scraping on entities and data preprocessing), dataset alignment that aligns collected sentences of different languages into parallel data, and dataset balancing that attempts to include a similar number of sentences for each category. In order to showcase the toolkit, the paper further presents two case studies on high-resource languages (i.e., English, Arabic, Spanish and Russian) and low-resource languages (Swahili-English). Implementation details of the toolkit on the cases are presented. Two datasets are hence collected and experimental results on the two collected datasets are reported with gender-oriented analysis. The major contributions of the paper are hence the presented toolkit for constructing gender-balanced parallel data and two case studies on the two collected datasets.

---

> ### Author Response · Authors · 2022-08-12
> **Pointing the limitations of our toolkit. Ongoing work on extracting more data for the 25th August.**
>
> Thanks for your comments!!!!
>
> W1. The sizes of the presented datasets for high/low-resource languages are small, which makes the sufficiency of using the collected data to evaluate gender bias in machine translation skeptical. It would be much better if the paper could discuss the size issue, e.g., which factor has a great impact of the final size of the collected dataset, whether the toolkit can be used to collect sufficient instances for languages that do not have many entities in Wikipedia.
>
> R1. This is a limitation of our tool. The size of the extracted data is limited to the amount of available data in Wikipedia biographies. We added this as a limitation in the Appendix section.
>
> W2. Although details of the toolkit are presented, OccGen seems to be quite sophisticated and manual postedition is required to build the final benchmark dataset, which could make it difficult to adapt the toolkit to other languages.
>
> R2. Our toolkit is already been used by other research teams, see: https://huggingface.co/datasets/projecte-aina/ca_zh_wikipedia
>
> W3. It is not clear how the work presented in the paper is suitable to this track which prioritizes datasets and benchmarks while the paper seems to focus on a toolkit. Intrinsic analysis on the collected datasets and clear definition of benchmark tasks on the collected datasets seem to be missing.
>
> R3. Since we have seen this as a limitation, we are currently using our toolkit to extract and release a bigger dataset that can be used as training data. We plan to release this balanced training dataset for the camera ready [ONGOING]. Also this work is suitable for the track, as it is mainly for creating datasets with special characteristics, so it can be used for creating datasets and benchmarks as the track is suggesting.
>
> W4. The paper states that it attempts to address both evaluation and mitigation issues. However, it is not clear how gender bias is mitigated in machine translation with a small collected balanced dataset.
>
> R4. We want to clarify this point. We are not currently mitigating gender bias, but if extracting balanced training data, general models can be fine-tuned with this data for this purpose. Temptatively, we can add these experiments for the camera ready version if we have time [TEMPTATIVE]
>
> W5. The analysis presented in experiments seems to be language-dependent. It is hence not clear whether the conclusion from this analysis could be extended to other languages.
>
> R5 Our analysis is dependent on language grammatical gender. In this sense, yes, if two languages express gender differently as English, Spanish, Arabic, Russian and Swahili do... the analysis will be different. Probably, the analysis between languages which express gender in a similar way (or only with minor lexical differences) i.e. Spanish and Catalan, will be similar.
>
> W6 The analysis on the more/less attention to the source sentence and words in the target sentence seems to be short of empirical evidence. Visualization or deep look into NMT models on gender generation could be used as the support to this conclusion.
>
> R6. As we mention in the conclusions, this is ongoing work.

---

> > ### Author Response · Authors · 2022-08-29
> > **Released more data.**
> >
> > Regarding R3, we have now released more data (1.5M sentences). See appendix G.

---

### Official Review · Reviewer_kgaf · 2022-07-25
**Contribution chooses to ignore non-binary people**

**Rating:** 3
**Confidence:** 4

**Strengths:**

The paper is very clearly written, and the motivation well outlined. The techniques used seem generalizable to other contexts.

**Weaknesses:**

The major weakness of this paper is related to their misrepresentations of gender and their misleading claims regarding gender diversity. The authors claim in the abstract that "OccGen can extract datasets that reflect gender diversity (beyond binary) more fairly in society." The authors repeat this in the introduction. However, in the actual article itself, the authors only use binary gender (and incorrectly call these "male and female" instead of man and woman.) The authors go on and state that they are "exempt" from any responsibility about this decision since it is out of scope.

This is honestly confusing to me because the authors' stated goal of this work is to address harms of gender bias related to allocation and representation. This contribution replicates harms it claims to be combating in its exclusion of non-binary people. This is especially significant as this is a datasets and benchmarks track. Have the authors considered how this (lack of) representation harms non-binary people? The authors made the decisions about the construction of the dataset. I do not see this decision as "out of scope" in any meaningful way. Non-binary people were even included in the data presented in Figure 2 (albeit in much smaller numbers), but the authors chose not to include any of them at all.

Of course, not every paper can be everything to everyone. But it is clear that the authors are familiar with the relevant literature on gender bias and gender diversity, having cited many articles I might have suggested. The authors had a real opportunity to engage gender bias in a meaningful way but instead chose to ignore it, call it "out of scope" and link out to "wonderful online resources" that they do not engage with. At the very least, the authors need to meaningfully engage with the limitations of their work with regard to use of binary gender over other representations. But, the nature of the work presented, even if it could be generalized and might otherwise have meaningful technical contributions, is grounded in binary gender and gender "balance" so I cannot support it in its current form.

In addition to the harms and ethical implications, a focus on binary gender limits the potential impact such a contribution could have. There could have been an interesting discussion on gender bias in each language, but that seems to have been reduced to a single sentence (279-280). There also could have been a discussion about how the technique could be meaningfully expanded beyond a representation of binary gender, or what kind of dataset requirements would be necessary to improve results for more/all genders.

**Additional Feedback:**

Reference to the Appendix is missing (??), line 76.

**Clarity:**

I think there is something missing in the sentence on lines 22-23. Maybe remove the "while"? Otherwise, the paper is clearly written.

**Correctness:**

When referring to binary gender, the terms "man/woman" are the correct terms to use, not "male/female." Of course, this does not account for other genders.

**Documentation:**

The datasets are in an easy to use CSV format. There seem to be several TODO items not completed on the Github page for the toolkit.



**Ethics:**

Per my discussion in the weaknesses section, I think this criterion 3 applies in its use of binary gender (while claiming to represent gender diversity):

> 3. Encode, contain, or potentially exacerbate bias against people of a certain gender, race, sexuality, or who have other protected characteristics. For instance, does the dataset represent the diversity of the community where the approach is intended to be deployed?

In particular, this research focuses on machine translation and is developed by researchers at Meta, suggesting that this technology may be used for social media purposes. Although with a quick search I could not find information about the percentage of non-binary people on Facebook, the number is undoubtedly non-zero. And Facebook itself introduced the option for people to identify as genders other than man and woman in 2014, 8 years ago. Like many marginalized groups, non-binary people are likely to rely on social media and the internet for connecting with others, even more than cisgender people. It is important that there is representation for them in machine translation and other contexts. The authors clearly have the capability to engage with this question, but chose not to, and also (incorrectly in my opinion), argue that doing so is out of scope.

**Relation To Prior Work:**

Yes.

**Summary And Contributions:**

In this article, researchers present a dataset and toolkit called OccGen, which uses a selection of entities from Wikidata to generate a "gender balanced" dataset that can be used for machine translation. The authors use their sample dataset to demonstrate differences in translation outputs based on binary gender.

---

> ### Author Response · Authors · 2022-08-12
> **Clarifications on our work**
>
> Thanks for raising discussion points!
>
> W1:The major weakness of this paper is related to their misrepresentations of gender and their misleading claims regarding gender diversity.
> …
> Non-binary people were even included in the data presented in Figure 2 (albeit in much smaller numbers), but the authors chose not to include any of them at all.
>
> RESPONSE:
> Our tool allows us to extract beyond binary gender. We limited our use cases to binary because just including a third gender from figure 2 (e.g. transgender female, the third most representative) would have resulted in less than 79 entities for en-sp-ar-ru and 0 entities for en-sw. Note that since our tool only extracts natural data, if we want it balanced, then the amount of data is limited to the amount of data available in the less representative gender. However, once Wikipedia starts to be more representative in gender, our tool will allow the community to extract such balanced data.
>
> W2: There also could have been a discussion about how the technique could be meaningfully expanded beyond a representation of binary gender, or what kind of dataset requirements would be necessary to improve results for more/all genders.
>
> RESPONSE: Based on Figure 2, using beyond binary gender in our use cases would have extremely reduced the released data to less than 79 entities (if including transgender female, the 3rd most representative gender) for en-es,ru,ar and to 5 in en-sw (if including non-binary, the 3rd most representative gender for this pair). Since we need at least hundreds of sentences to evaluate, we could unfortunately not use beyond binary gender. However, for other languages or when Wikipedia is more gender representative, OccGen can provide balanced data on gender within occupations beyond binary gender.
>
> D1: When referring to binary gender, the terms "man/woman" are the correct terms to use, not "male/female." Of course, this does not account for other genders.
>
> RESPONSE: We use the labels used in the source data that we are using: Wikipedia. From a strictly grammatical point of view, the only thing that can be said is that "male, female" can used both as nouns or adjectives, while "man, woman" are mainly used as nouns. From a lexicographical point of view, "male, female" typically refer more specifically (although not only) to the biological domain. The point here is that in the field of gender diversity, the notion of gender is different from the notion of biological sex. Our paper assumes this view of gender identity, and it would make sense to use terms that are currently used to refer to gender (such as "man, woman") rather than terms that refer to sex (such as "female, male"). However, we use the labels from our source data for coherence.
>
> E1: In particular, this research focuses on machine translation and is developed by researchers at Meta, suggesting that this technology may be used for social media purposes. Although with a quick search I could not find information about the percentage of non-binary people on Facebook, the number is undoubtedly non-zero. And Facebook itself introduced the option for people to identify as genders other than man and woman in 2014, 8 years ago. Like many marginalized groups, non-binary people are likely to rely on social media and the internet for connecting with others, even more than cisgender people.
>
> RESPONSE: Note the variety of institutions in the authors: Meta AI, Universitat Politècnica de Catalunya and Batou XYZ. The tool does not use social media data at any step.  OccGen is a Wikipedia-based tool.

---

> ### Public Comment · ~Pascale_Fung1 · 2022-08-30
> **We should not stop social progress**
>
> https://openreview.net/forum?id=tTPVefaATp6&noteId=9PEkaMNq1X

---

> ### Comment · Reviewer_kgaf · 2022-09-01
> **Comment on my final review**
>
> Thank you to the authors, other technical reviewers, ethics reviewer, and additional commenters for the engaging discussion about this paper. While I appreciate that the authors have made improvements to their paper and the dataset as the discussion has unfolded, I do not think that these changes are sufficient for me to recommend acceptance, and I will not be changing my initial recommendation. As it currently stands, I don't think the paper is ready for publication, but I could see steps that could be taken for it to be published in the future. I am sharing some suggestions as a path forward. Of course, the final decision lies with the AC/PCs.
>
> I think that an improvement to the dataset and/or the method is required for acceptance. For example, if the dataset were to include real or synthetic entries that enabled it to be used for genders besides man and woman. Or, a different method could be used that would not require "gender balance," thus enabling improvements in gender diversity with smaller and/or unbalanced datasets. While, as the authors point out, the dataset and associated methods represent advancement in the field for their inclusion of women, this is a short-term advancement that could have negative long-term implications. Once a dataset is released, the field is likely to use said dataset rather than working to release datasets that are more inclusive. Additionally, advancing methods that require "gender balance" likewise entrench the gender binary, even if theoretically the methods could operate outside it. Additional work is therefore required for the potential strengths of the authors' methods to be realized, and I hope that the authors consider the critiques presented here and continue working on this important problem.

---

> > ### Author Response · Authors · 2022-09-01
> > **The added training data includes real entries for genders besides man and woman, please check conclusions + appendix G**
> >
> > Thanks for your reply.
> >
> > However, we would appreciate if you can check the new version of the paper which includes a new data set that can be used for training and which includes gender categories beyond man and woman. The data set is available in our github: https://github.com/mt-upc/OccGen_dataset

---

> > > ### Comment · Reviewer_kgaf · 2022-09-01
> > > **Clarification on prior comments**
> > >
> > > I apologize for not being more clear in my comments above. I understand that the authors have provided a larger dataset, and that this technically means the dataset could be used for genders besides man and woman as I said in my comment. I should have been more clear, but what I mean is for these categories to be used in any functional or practical manner. By the authors' own admission: "This means that a parallel dataset in four languages, and balanced in all genders with non-zero entities from Wikipedia sources, would be very very small (with 1 or 2 entities in total). " and "We suggest that this data can be balanced by means of artificial techniques such as oversampling, counterfactual techniques or other synthetic techniques in future." I don't think just releasing the whole corpus addresses the underlying concerns, as it defers the responsibility to future users of said dataset to develop their own techniques for gender inclusiveness, rather than considering them prior to acceptance of the present work.
> > >
> > > Also, to consider the dataset more closely, it has a number of issues even besides the differences in numbers. For the paper to be accepted, this categorization would need to be completely reworked. The categorization of gender groups provided by Wikipedia don't make sense; as I've said several times, male/female are not appropriate terms for gender. Additionally, "transgender female" is not a gender. Trans women are women. I don't understand why there needs to be a separate category here. I do not understand how this new corpus could be meaningfully used to address the gender bias/diversity questions raised by the paper without significant additional work.
> > >
> > > Additionally, even with the release of a larger dataset, no additional analysis has been conducted (from what I can tell in the PDF, please correct me if I'm mistaken), and the paper continues to misrepresent its capabilities to go "beyond the binary," and continues to define any gender categorization as out of scope. And there are likely additional concerns I've discussed in previous comments to be reviewed. This contribution feels like it is on its way toward an inclusive contribution, but just isn't there yet in my opinion.

---

> > > > ### Author Response · Authors · 2022-09-02
> > > > **Clarification on the contribution of the paper**
> > > >
> > > > Thanks for your clarification. We understand your complain about the categorization of gender groups used by Wikipedia. However, this is not the focus of the paper. We do not pretend to provide a categorization of gender at any point. We use the perceived gender which is explicitly annotated in Wikipedia.
> > > >
> > > > We explicitly recognise the limitations of our work and explicitly mention them in the paper as you cite. However, our tool and our new corpora are the first that considers categories beyond binary gender. This tool and our corpora can be used to explicitly tackle the representations of different genders in NLP. For this we consider it a big step forward.

---

### Official Review · Reviewer_bq7Q · 2022-07-26
**A timely dataset for gender and occupation concerning multilingualism and gender balance**

**Rating:** 7
**Confidence:** 4

**Strengths:**

- Occupation is an important attribute across languages that concern gender balance since the output trained with or analyzed by such balance may show unwanted inductive biases that harms e.g., the end users of products
- Authors select high and low resource language families and the representatives to check how the pipeline works and how the result comes out with such languages

**Weaknesses:**

- Some questionable approach regarding multilingual alignment (detail in Correctness section)
- It seems that checking only BLEU may not highlight the utility of the proposed method (detail in Prior Work section), and the analysis of the evaluation result is weak; there is little to learn except that the performance of mBART is significantly inferior compared to other candidates

**Additional Feedback:**

Comments

- One of my concerns is that, though authors claim that the proposed extraction scheme is suitable for gender-balanced corpus that regards various aspects of gender, but it is currently a claim and is not shown by the evaluation. I wonder if there is any evaluation methodology that helps the users of the proposed method can see if the corpus is well extracted regarding various gender spectrum in Figure 2.

- Next, to be honest, Discussion section (especially the first paragraph) is difficult to understand due to its conceptual usage of terms and it would be much better if there are some example sentences of improvement regarding some of the languages. Especially, what does the author mean by line 263: 'English and Spanish have a solid language generation'? if it is due to the script-level similarity with the source language (English), doesn't it hide the effectiveness of the proposed evaluation methodology?  Also, is there any authors' suggestion on line 287?

- Last, this is just for a suggestion; I think it may be nice to tell readers how to pronounce the datset; e.g., 'oxygen'?

Addiotionally, I suggest some typos or missing contents here.

- I think citation formats are all with authors' names, which makes it difficult for readers to tell the citation form the text body.
- '. While' to ', while' in line 22
- line 68; where is Table 2?
- Appendix ?? in line 75

**Clarity:**

This paper is well-written and has a nice structure, but some typos and missing contents are observed.

**Correctness:**

It seems that the claims are correct overall, and the resulting dataset seems to be constructed in a sound way with an appropriate evaluation method, but some approaches seem sub-optimal. For instance, in multilingual alignment, using pivot language is an intuitive approach that makes the problem much easier, but I think authors need to explain more on using such approach; if the number of languages that are used are not large enough, why is not alignment made in pairwise way? This does not necessarily harm the quality of the paper or other claims, but I think it regards correctness of the claims.

**Documentation:**

There seems to be many todos in the github at this point. Documentation need proper updated upon the acceptance.

**Ethics:**

The paper describes substantially on the necessity of the proposed scheme in ethical viewpoint, and the data card is appropriately suggested as a supplementary material.

**Relation To Prior Work:**

There seems some missing works in the evaluation using machine translation. Using only BLEU may not shed light to the effectiveness of the proposed extraction scheme. Does the level of BLEU indicates the gender-unbiasedness of the translation result? I think, to check the balancedness of the augmented multilingual dataset, one would have to fine-tune a translation model with the augmented dataset and check the model correctly or unbiasedly infer some occupation-related statements suggested in previous work on translation gender bias (I think the authors may be well aware of them). Since this is just a suggestion and may go beyond the scope of the proposed dataset extraction scheme, so I do not see this as a limitation; however, if authors decided to use BLEU as the only evaluation metric of MT, I suggest the addition of why not using other fairness-related metrics.

**Summary And Contributions:**

This paper suggests OCCGEN for the extraction of multilingual parallel data balanced in gender within occupations.

---

> ### Author Response · Authors · 2022-08-12
> **Clarifying the motivation of multilingual alignment. Currently working on a human analysis [aiming at 25th August].**
>
> Thanks for your comments!
>
> W1. It seems that checking only BLEU may not highlight the utility of the proposed method (detail in Prior Work section), and the analysis of the evaluation result is weak; there is little to learn except that the performance of mBART is significantly inferior compared to other candidates
>
> RESPONSE 1. We are preparing a human evaluation that will give more insights. We are annotating the occupations and pronouns and we will check the quality of the translation for each several eval sets. [ONGOING]
>
> C1. For instance, in multilingual alignment, using pivot language is an intuitive approach that makes the problem much easier, but I think authors need to explain more on using such approach; if the number of languages that are used are not large enough, why is not alignment made in pairwise way? This does not necessarily harm the quality of the paper or other claims, but I think it regards correctness of the claims.
>
> RESPONSE 2. One of the strengths of the tool is that it can be highly multilingual, and we have designed it to be optimal for this. The pairwise way can always be used by using 2 languages at a time.
>
> RPW1.  I think, to check the balancedness of the augmented multilingual dataset, one would have to fine-tune a translation model with the augmented dataset and check the model correctly or unbiasedly infer some occupation-related statements suggested in previous work on translation gender bias (I think the authors may be well aware of them). Since this is just a suggestion and may go beyond the scope of the proposed dataset extraction scheme, so I do not see this as a limitation; however, if authors decided to use BLEU as the only evaluation metric of MT, I suggest the addition of why not using other fairness-related metrics
>
> RESPONSE 3. We are currently extracting balanced training data for several languages. If time allows (deadline of 25th August for the camera ready), we can use this new data to fine-tune the M2M model (which is one of the best performing models) and add conclusions in the final version. [TEMPTATIVE FOR THE CAMERA READY]

---

> > ### Author Response · Authors · 2022-08-29
> > **Human evaluation included**
> >
> > As an update to response 1, details of this human evaluation are in the last paragraph of section 5.2 and appendix D.

---

### Official Review · Reviewer_a16m · 2022-07-28
**Great work on building the parallel dataset balanced in gender within occupations.**

**Rating:** 6
**Confidence:** 4
**Correctness:** 1. The balancing method is sound, and…
**Clarity:** 1. Line 25

**Strengths:**

1. **Significance**: Compared to other work, the released toolkit aims to extract real-world data, ensuring the reliability of data sources. Besides, it is customizable for different gender/occupation options, which is considerable for non-binary gender.
2. **Accessibility**: Code and data are hosted in the open-source platform and can be accessed instantly.
3. **Implications**: The released toolkit can be used to extract additional multilingual parallel data without the above-mentioned bias. The community may directly utilize the released benchmarks to analyze MT behavior in terms of binary gender. With customized settings, it also can help the researcher in the field of MT to create more benchmarks to study the other bias of MT models.


**Weaknesses:**

1. **Relevance to the broader research community**: This work only evaluates the performance of pre-trained MT models using the released benchmarks. It should also study diverse MT models including vanilla Transformer to extend the scope of experiments.
2. **Contribution**: The contribution of this work is a little bit limited. The proposed pipeline highly relies on the existing methods (collection and alignment). From the perspective of methodology, it only contributes a customizable balancing algorithm.

**Additional Feedback:**

1. BLEU score is a general-purpose evaluation metric. It would be better to use a metric like F-score to evaluate the bias in the analysis part, focusing on the translation of special terms related to gender.
2. According to the submission instructions, there should be additional pages containing the paper checklist (https://neurips.cc/Conferences/2022/CallForDatasetsBenchmarks).

**Documentation:**

The author provides a URL for reviewer access to the dataset. But the documentation of this toolkit is still under construction during the reviewing period.

**Ethics:**

Potential ethics discussion: The released datasets are collected from Wikipedia. Is there copyright problem or personal information?

----------
***Update:***  Gender diversity is indeed important when it comes to the non-binary gender. The authors have made some clarifications during the rebuttal period. There are few records of non-binary people in Wikipedia source, so the experiments/results of gender bias were presented in a binary way. However, the toolkit still provides the customizable options to extract data for non-binary gender, as I mentioned in the initial review (Strength [1]).

**Relation To Prior Work:**

The related papers are cited and dicussed.

**Summary And Contributions:**

This paper contributes a toolkit that is capable of extracting multilingual parallel data balanced in gender within occupations from real-world data. Specifically, the pipeline of this toolkit consists of three main stages: data collection, datasets alignment, and dataset balancing, where the balancing methodology is customizable in both gender and occupations. To show the effectiveness of the proposed methodology, the authors create the evaluation benchmarks of machine translation (MT) for 4 high-resource languages (Arabic, English, Russian, Spanish) and one low-resource language (Swahili-English). Besides, it conducts experiments and discusses the translation performance of different languages in terms of gender.
To conclude, the contributions of this work are:
1. Proposing a methodology that can create multilingual parallel data balanced in gender within occupations.
2. Contributing five English-centric parallel datasets for both high-resource and low-resources languages, which are created by the proposed balancing methodology.
3. Provide a preliminary analysis of three pre-trained MT models regarding the translation behaviors of gender bias.

---

> ### Author Response · Authors · 2022-08-12
> **Documentation ready, clarifying contribution and answers to some other questions.  Currently working on human analysis for the 25th August version.**
>
> Thank you for pointing out the weaknesses of the paper.
>
> W1. Relevance to the broader research community: This work only evaluates the performance of pre-trained MT models using the released benchmarks. It should also study diverse MT models including vanilla Transformer to extend the scope of experiments.
>
> RESPONSE-1: We focused on experimenting with existing Multilingual Systems because they tend to outperform the bilingual ones. Also we want the experiments to be mainly handled in multilingual settings, and to have comparative study between the HR and the LR, we used the same multilingual setting for both cases
>
> W2. Contribution: The contribution of this work is a little bit limited. The proposed pipeline highly relies on the existing methods (collection and alignment). From the perspective of methodology, it only contributes a customizable balancing algorithm.
>
> RESPONSE-2: This customizable balancing algorithm simplifies the way of getting balanced data. The purpose of OccGen is that when using Wikipedia Data, it can be used as balanced data in a multilingual setting. We consider this a great contribution, not only does it facilitate creating balanced data in a multilingual setting, but also enables the user to use the toolkit to have unbalanced data whether parallel, or multilingual. Given that, our toolkit simplifies the extraction of parallel data, it is already been used for other research groups see this data that was extracted using the OccGen toolkit: https://huggingface.co/datasets/projecte-aina/ca_zh_wikipedia
>
> D1. The author provides a URL for reviewer access to the dataset. But the documentation of this toolkit is still under construction during the reviewing period.
>
> RESPONSE-3: The documentation is being updated now, the dataset is available at https://github.com/mt-upc/OccGen_dataset and the toolkit is available at https://github.com/mt-upc/OccGen_toolkit
>
> E1. Potential ethics discussion: The released datasets are collected from Wikipedia. Is there copyright problem or personal information?
>
> RESPONSE-4
> As informed in Wikipedia: Most of Wikipedia's text is co-licensed under the Creative Commons Attribution-ShareAlike 3.0 Unported License (CC BY-SA) and the GNU Free Documentation License (GFDL) (unversioned, with no invariant sections, front-cover texts, or back-cover texts). Some text has been imported only under CC BY-SA and CC BY-SA-compatible license and cannot be reused under GFDL.
>
> AF1. BLEU score is a general-purpose evaluation metric. It would be better to use a metric like F-score to evaluate the bias in the analysis part, focusing on the translation of special terms related to gender.
>
> RESPONSE-AF1 We are working on a human evaluation which consists in annotating the occupations/pronouns in our evaluation data sets and labelling the quality of the translation.  [ONGOING]
>
> AF2. According to the submission instructions, there should be additional pages containing the paper checklist (https://neurips.cc/Conferences/2022/CallForDatasetsBenchmarks).
>
> RESPONSE-AF2: We have added it now. Thanks for noticing it!

---

> > ### Comment · Reviewer_a16m · 2022-08-21
> > **Reply to authors' response**
> >
> > Thank you for your reply.
> >
> > - RESPONSE-3: I have checked the updated documents. It covers most parts of usage regarding the toolkit and open-sourced data.
> > - RESPONSE-4: Since most of Wikipedia's text is licensed, the legality and privacy may not be the potential problems. It would be better if the committee could conduct an ethics review for the open-sourced data.
> > - RESPONSE-AF1: I am looking forward to seeing the results of the human evaluation. But is there any automated metric for evaluating it?
> > - RESPONSE-AF2: Thank you for adding the checklist. Now current version of the paper follows the guideline of this track.

---

### Official Review · Reviewer_h6dn · 2022-07-28
**More Data Please!**

**Rating:** 5
**Confidence:** 4
**Correctness:** Yes.
**Clarity:** Good.

**Strengths:**

1. The paper topic is very important.
2. The paper describes the OCCGEN toolkit, which generates multilingual balanced datasets in gender within occupations.
3. The experiment result shows the multilingual machine translation models perform better on male entity samples, which is intuitive.


**Weaknesses:**

***Main Weakness***
0. This paper presents the OccGen toolkit that builds multilingual parallel data sets balanced in gender within occupations. However, this track is ***Datasets and Benchmarks***, so why not just release the refined multilingual parallel data sets to the audiences? The current dataset is too tiny, which only consists of nearly 1K sentences . Furthermore, all the experiments and analyses are conducted with the tiny dataset, which can not well support the paper's claim.

The other weaknesses:
1. The proposed gender balancing algorithm performs gender balancing through data selection, which operates well in binary gender. However, it can not handle the minority genders. For instance, in Figure 2, if we consider the ‘transgender female,’ according to the gender-balance hypothesis, much data from male and female entities could be discarded.
2. Similarly to the main weakness, the experiment is inconsistent with the motivation. In Line 21, the motivation is to create a balanced dataset for training and evaluation. However, the experiment section fails to show the proposed balance dataset’s effectiveness in reducing the gender bias of the translation model.

Overall, I would like to see directly released training data.


**Additional Feedback:**

1. How does OCCGEN help to refine the training data?
2. What's the performance of the translation model trained with the refined data? Will the overall translation quality decrease?

**Documentation:**

Yes, but the provided data is limited.

**Ethics:**

No.

**Relation To Prior Work:**

Yes.

**Summary And Contributions:**

This paper describes the OCCGEN toolkit, which extracts gender-balanced multilingual parallel data, and evaluates three multilingual models in high-resource and low-resource gender-balanced datasets. The proposed toolkit extracts entities of all the occupations from wiki data and obtains the corresponding gender and text information. Furthermore, it acquires parallel sentences through sentence embedding from the multilingual sentence encoder and parallel sentence alignment following the margin-based criterion. Finally, the dataset balancing algorithm generates the gender-balanced dataset through entity categorisation and gender balancing within occupations. The authors report two use-cases of the OCCGEN toolkit on high-resource languages and low-resource languages. Experiment results show that the multilingual machine translation models perform better on male entity samples.

---

> ### Author Response · Authors · 2022-08-12
> **Extracting more data to add [working on it for the 25th August]**
>
> Thanks for pointing out the weaknesses of our paper, we are currently working on how to address them as follows:
>
> W0: This paper presents the OccGen toolkit that builds multilingual parallel data sets balanced in gender within occupations. However, this track is Datasets and Benchmarks, so why not just release the refined multilingual parallel data sets to the audiences? The current dataset is too tiny, which only consists of nearly 1K sentences . Furthermore, all the experiments and analyses are conducted with the tiny dataset, which can not well support the paper's claim.
>
> RESPONSE-0. We are using OccGen toolkit to extract training data for En-Es, En-Ar and En-Ru. Extracting entities for the parallel language pairs is different from the high resource case as each language pair would need extracting a different set of entities from Wikipedia and apply the full process on each pair individually.
> For the case of En-Sw we are re-running the toolkit to see if there is more available data in this case. We should have this data for the camera ready version. [ONGOING]
>
> W1: The proposed gender balancing algorithm performs gender balancing through data selection, which operates well in binary gender. However, it can not handle the minority genders. For instance, in Figure 2, if we consider the ‘transgender female,’ according to the gender-balance hypothesis, much data from male and female entities could be discarded.
>
>
> RESPONSE-1. The reviewer points out that “ much data from male and female entities could be discarded when adding transgender female data“.  This is a consequence of the fact that our method extracts balanced data from natural data: our sets can not surpass the number of entities of the less representative gender that we want to include.
> However, we want to point out that extracting balanced data which does only contain natural data is also one of our strengths. In all cases, we are providing only natural data and not synthetic data. Therefore, our set does not contain synthetic data artificially created with data augmentation and this avoids all the counterparts that synthetic data has.
>
> W2. Similarly to the main weakness, the experiment is inconsistent with the motivation. In Line 21, the motivation is to create a balanced dataset for training and evaluation. However, the experiment section fails to show the proposed balance dataset’s effectiveness in reducing the gender bias of the translation model.
>
> RESPONSE-2. Once we extract the training data, if we have time before the 25th August, we can use it to fine-tune the current systems. In case we can make it, we plan to do that for M2M model since it is one of the best performing systems and available for all languages that we have used. [TEMPATIVE FOR THE CAMERA READY]
>
> Additional Feedback (AF): How does OCCGEN help to refine the training data? What's the performance of the translation model trained with the refined data? Will the overall translation quality decrease?
>
> RESPONSE- AF. Additionally feedback will be addressed with responses 0 and 2.

---

> > ### Author Response · Authors · 2022-08-29
> > **More data is available now**
> >
> > Regarding Response-0, we are now releasing around 1.5M sentences of bilingual data for each of the following pairs of languages: English-Spanish (En-Es), English-Arabic (En-Ar) and English-Russian (En-Ru). This new data set includes all the gender categories with non-zero entities from the Wikipedia sources in these languages, as shown in Figure 2. However, this set of additional data now does not have the same number of entities in all gender categories.  We suggest that this data can be balanced by means of artificial techniques such as oversampling, counterfactual techniques or other synthetic techniques in future.

---

### Review · Ethics_Reviewer_ffcn · 2022-08-26

**Recommendation:** 3

**Ethics Review:**

## Note:

*The original ethics review was posted as an 'official comment', because there was nowhere to submit an ethics review on the platform. I am copying the original comment as an ethics review, along with the response that the authors gave to that review, in case it gets lost in the transfer.*

## Ethics Review

The paper proposes the OCCGEN toolkit to generate monolingual, bilingual, and multilingual balanced datasets in gender within occupations.

Reviewer kgaf has already summarised one of (what I take to be) the key weaknesses and ethical issues with this project: namely, that the authors effectively ignore non-binary gender. This essentially serves to further marginalise—and will likely perpetuate data-driven biases against—already-marginalised groups, which is orthogonal to the stated goal of the paper to to address harms of gender bias related to allocation and representation with respect to these data.

It does not strike me as sufficient to pay lip service to discrimination or bias against non binary individuals while at the same time further entrenching the dominant paradigm within the field—i.e., the emphasis of binary gender in such datasets. For example, the authors say that ‘OCCGEN is customizable in gender categories considering the broad gender spectrum and balanced within occupations; such features help be more inclusive and mitigate stereotypes’ but they do not themselves do this.

The general ethical conduct section of the ethical review guidelines on there NeurIPS website clearly states that ‘submissions must adhere to ethical standards for responsible research practice and due diligence in the conduct’, and if the authors use human-derived data (which the present authors do, since their data are derived from Wikipedia. As such, the authors are to consider whether these data might ‘encode, contain, or potentially exacerbate bias against people of a certain gender’. It seems fairly obvious in this case that the proposal runs afoul of this standard.

Using the authors’ own definition of bias—as ‘one relevant factor that prevents our systems from being equitable’, it should be apparent that the proposal is inherently biased against non-binary individuals. In short, it seems false that these considerations are beyond the scope of this paper, as the authors try to suggest.

## Authors' Response

The authors had responded to the above as follows:

> Our tool can extend to non-binary case, including all gender categories from Wikipedia. Given the current limitation of Wikipedia in representing genders beyond male, our data focuses on binary gender (man/woman) in order to release more data. We have to keep in mind that at the moment our models are biased towards the most prevalent gender which is male. We are releasing data for which we are mitigating this prevalence and balancing male and female genders. Moreover, we are releasing a tool which can be easily extended to include more genders from Wikipedia, once there exists enough entities related to this gender. Once, we have these entities from Wikipedia, we can further mitigate the male prevalence to include all genders categorized in Wikipedia, improving inclusiveness. We are not neglecting the other genders, we just could not find enough entities representing them from Wikipedia. Already the research community is suffering from lack of data representing females and we are solving this for now. We are not aware of another open-source biography platform which includes other genders to a greater extent. Ours is the first big step towards doing so.

This is response is insufficient to offset the ethical considerations noted above, and by some of the other reviewers. In particular, Ben Green (2019, ['"Good" Isn't Good Enough'](https://www.benzevgreen.com/wp-content/uploads/2019/11/19-ai4sg.pdf)) points out that incremental "good", of the sort that is proposed by the authors in their response, can lead to long-term harm. Their response is precisely the sort of step-wise movement of which Green is critical.

---

> ### Author Response · Authors · 2022-08-29
> **More data and more genders**
>
> While our tool enables us to extract balanced natural data of all genders, in practice the amount of data is lower bounded by the amount of the minority gender data. This means that a parallel dataset in four languages, and balanced in all genders with non-zero entities from Wikipedia sources, would be very very small (with 1 or 2 entities in total).  This is a reflection of societal bias in the natural data. Meanwhile, our proposed multilingual dataset with binary gender (women/men) balance is an important  first step on the path of progress away from a male-dominated gender bias.
>
> Nevertheless, to address the concern of the reviewers regarding the small size of the data and with only binary gender balance, we are now releasing around 1.5M sentences of bilingual data for each of the following pairs of languages: English-Spanish (En-Es), English-Arabic (En-Ar) and English-Russian (En-Ru). This new data set includes all the gender categories with non-zero entities from the Wikipedia sources in these languages, as shown in Figure 2. However, this set of additional data now does not have the same number of entities in all gender categories.  We suggest that this data can be balanced by means of artificial techniques such as oversampling, counterfactual techniques or other synthetic techniques in future.

---

> ### Public Comment · ~Pascale_Fung1 · 2022-08-29
> **Ethics review should not stop social progress**
>
> I find this discussion extremely important. First of all, what is the purpose of an ethics review? It is to ensure that the kind of research published does not lead to intentional or unintentional harm. It is not to ensure that every research paper has done "sufficiently good".
>
> Looking at this paper, which proposes multilingual parallel datasets balanced in binary gender from Wikipedia, as a first step towards offsetting the usual male-centric bias in data for AI models, is an important and as shown, feasible way to mitigate harm towards binary women. I strongly disagree that by doing so it adds additional harm to other non-binary genders. In terms of social justice, progress is progress. Black men gained voting rights long before women ever did - they were both progressive steps towards equal rights. One does not negate the other. The authors have shown that data on non-binary gender is very rare in Wikipedia data, which represents an obvious societal bias. By illustrating how to balance the dataset for binary gender women, the paper shows the way towards more balanced corpus in the future for other gender groups. To say that this work is not sufficiently good therefore it discriminates against other gender groups is disingenuous,  and by rejecting such a paper out right, this review in fact stands in the way of social progress, not enables it.
>
> Regarding Ben Green's "opinion piece" (as it is not really an empirically verified scientific paper), we can also have another debate as to whether "Good isn't good enough" is in itself a good enough reason to stop AI researchers from doing work on social progress? I think we should encourage all work, and collectively make more progress together.  (The argument that computer scientists are techno-centric is a circular argument. We built the systems, so we need to build solutions to the problems we created in the systems. But this is a digression to the discussion of this paper.) ( To accept the opinion of that of a man to reject a paper on gender balance is also a problem, not the solution.)
>
> In conclusion, 1) "canceling" feminism - known as the fight for equal rights for (biological? binary?) women - does not enable the fight for other mistreated groups; 2) rejecting papers because the work is perceived to be not having done "sufficient good" is not the purpose of ethics review. We should encourage all work on social progress, not dismiss any because it is not doing all the good. 3) research in nature is never "good enough". Every paper presents work that is hopefully a progress on prior work.

---

> > ### Public Comment · ~Kendra_Albert1 · 2022-08-30
> > **Response to Professor Fung**
> >
> > As an expert in gender and machine learning in the context of nonbinary people, I am grateful the ethics reviewer took their responsibility seriously and I agree with their assessment. However, I felt moved to comment on this paper because of the response from Professor Fung, which manages to both be offensive and assert a number of highly contested claims as true without evidence.
> >
> > Many of the arguments that Professor Fung raises re: “progress” and incremental improvements are addressed by the Green piece itself, which I will note, was accepted in a workshop associated with NeurIPS. But the example she uses re: voting is actually a perfect one to illustrate why her claims about progress do not reflect any kind of consensus. Assuming Professor Fung is referring to the United States, the idea Black men got the vote before women  is technically correct but functionally untrue. Post Reconstruction, in most parts of the South, the existence of Jim Crow laws effectively prevented Black men from voting prior to the Civil Rights Act. But more on topic, the (racist) argument that Black men did not deserve to vote before White women was a key argument by suffragettes including Elizabeth Cady Stanton. So even in the example that Professor Fung cites for its obviousness, there was significant contemporary disagreement about whether a particular step was “progressive [] towards civil rights.”
> >
> > Of course, saying one’s position was adopted by noted racist Elizabeth Cady Stanton is not much of an endorsement, so I want to cite a few other examples of disagreements about the nature of "progress" in social justice. One might look at the considerable debate on the introduction of ENDA in the 1990s, when certain segment of the US gay rights movement endorsed a employment non-discrimination bill that excluded trans people under the theory that it was more likely to pass, causing significant backlash. [My own work](https://papers.ssrn.com/sol3/papers.cfm?abstract_id=4167103) explores this tension as well, arguing that technological reform solutions retrench certain forms of violence. The idea that we should encourage all steps towards “social progress” has been robustly critiqued, and it is very reasonable for a NeurIPS ethics reviewer to have assessed the circumstances and determined that such an incrementalist approach was not appropriate.
> >
> > Moving to the specifics of the paper, if Professor Fung and/or the authors of this paper believe that women are harmed by being underrepresented in datasets, it seems impossible to argue that [nonbinary people are not harmed by a dataset that excludes them](https://papers.ssrn.com/sol3/papers.cfm?abstract_id=4122529). If constructing datasets in ways that systemically cause gender bias is harmful, the authors should know better than to do it again. Just because the harm is methodologically convenient doesn’t make it not real. If the argument is that this dataset does not create additional harm to nonbinary people, I do not think Professor Fung or the authors would argue that datasets that claim gender balance but exclude women are appropriate, so one would expect that this dataset not make a similar claim with regards to nonbinary people.
> >
> > To the point about the appropriateness of this concern being expressed through ethics review, it appears that the authors basically ignored reviewer critiques about the dataset’s treatment of nonbinary people until it looked like the dataset would not be accepted due to the issue, at which point they magically found it within themselves to modify their methods and produce a partial solution. That suggests that, contrary to Professor Fung’s statements, the ethical review did its job of actually making sure that authors paid attention to harm to a marginalized group and taking steps to mitigate it.
> >
> > Finally, I want to turn to the rhetoric deployed in this review response. Professor Fung's statements that enforcing the ethical guidelines against harm to nonbinary people with regards to this work is "cancelling feminism" and that a citation to a work by a man in a review is somehow "accepting his opinion to reject a paper on gender balance" are patently ridiculous and incendiary. And her use of the phrase "biological women" to define who feminism is “for” is an indication of her lack of knowledge with regards to nonbinary and transgender people, not to mention the contested histories of feminism(s). The fact that she has previously served as the Diversity, Equity, and Accessibility chair for this conference makes these rhetorical positions and this lack of knowledge more embarrassing, but we all have spots we are stronger on and weaker on. I hope Professor Fung can commit to learning more about trans and nonbinary experiences, as well as the history and critiques of linear beliefs about “progress’ and incrementalism, and using that knowledge to better inform future work that she may decide to weigh in on.

---

> > > ### Public Comment · ~Pascale_Fung1 · 2022-08-31
> > > **Please read the paper and its ethics review**
> > >
> > > I welcome this debate and discussions. It is only by openly stating our differences in opinions that we can learn and progress.
> > >
> > > But I feel that we are distracting from the paper being reviewed. Please read the paper and the authors' responses - they have taken the reviewer comments into account and have extracted data of non-binary gender from Wikipedia and suggested potential ways to balance for all gender groups.  That is another progress made.  (Meanwhile, the reason they didn't also became apparent - there is simply not enough natural data from Wikipedia on nonbinary gender. )
> > >
> > > For our debate on feminism and social progress, we should have a panels or workshops to address the issue as pertain to our research field, which is that "what is the role of an ethics reviewer"? Is ethics review of research papers supposed to be constructive or punitive. Is the goal to encourage more papers on responsible AI? Or to punish the papers that are not "good enough"? Research is never good enough by definition.

---

> > > ### Public Comment · ~Pascale_Fung1 · 2022-08-31
> > > **Diversity AND Inclusivity**
> > >
> > > I want to address the topic of "diversity and inclusivity" head on. It is precisely about INCLUDING all types of people in the society and welcome the DIVERSITY of our backgrounds and work.
> > >
> > > The principle of diversity and inclusivity entails that we should do more work that includes the groups that are traditionally repressed - not to exclude that of any group. To comment on a paper to encourage the authors to include more gender groups is constructive. To reject a paper because it was about gender balance for binary groups is punitive and detrimental to the progress in the field.
> > >
> > > There is absolutely no reason to exclude any one and we all should be inclusive  - we are all should be feminists, and fight for the rights of oppressed groups.
> > >
> > > What I find problematic is the non-inclusive nature of how we measure progress.

---

> > > ### Public Comment · ~Pascale_Fung1 · 2022-08-31
> > > **Weighing In**
> > >
> > > I feel compelled to disclaim that I do NOT believe feminism is for "biological" women only. It is for the rights of women, in their social and identities as women - equal pay, access to education, reproductive rights, marriage rights, divorce rights, equal representation in politics, etc. etc. Feminism fights for the equal rights of a person who has been discriminated against because she is a woman, biological or not, period. LGBTQ rights are for people who are in the LGBTQ group, who are discriminated against because of these identities. There are people who are women and LGBTQ, and racial minority. They have been discriminated against in all these contexts. We don't need to be a lawyer or social scientists to have a voice. We can all "weigh in" with our experiences and opinions.  It is not a competition about who is more discriminated against, it is about working together for social progress. In the context of research, we must be constructive in our criticism.

---

### Meta-Review · Area_Chair_L6Xp · 2022-09-12

**Recommendation:** Accept
**Confidence:** 3

**Metareview:**

This paper describes a toolkit for building multilingual datasets that are balanced across preselected gender categories in occupations, drawing from source datasets such as Wikipedia. The toolkit performs data collection, aligns data, and balances data in a customizable way. Case studies are performed on a machine translation task for five languages, and translation performance is discussed in terms of gender identity.

We appreciate the engaged discussion by reviewers and authors for this paper. The topic under consideration is in an important area of study, and it is clear substantial effort has gone into the paper and underlying work. However, given that the goal of the paper is to address gender measurement, it is crucial that questions of non-binary gender identification be correctly addressed. The authors need to be clear and direct about the limitations of their work and the limitations imposed by the use of a dataset with an almost entirely binary selection of genders, rather than simply describing non-binary gender questions as “out of scope.” Additionally, the paper needs to be written to use preferred terminology with respect to gender in all areas.

This paper is being recommended for a *conditional acceptance*. In order for this paper to be accepted, the final camera-ready version of this paper will be checked to ensure that it:

- More appropriately represents the toolkit and dataset by removing inappropriate claims as to the inclusiveness of the dataset.
- Discusses the limitations of the presented work as it relates to the under-representation of non-binary people, and by explicitly engaging in the related cited work.
- Acknowledges the limitations of the gender taxonomy being inherited from Wikipedia.
- Ensures that all references to gender use the correct terminology.

Strengths:

- The paper is in a very important research area and is clearly written.
- The multilingual machine translation experiments demonstrate both the effectiveness of the toolkit in high- and low-resource languages, and the importance of gender in the task.
- Code and data have been made available for the community.

Weaknesses:

- The paper needs to handle questions of gender in a more inclusive way, lest it perpetuate harms associated with restricting gender to a binary.
- The experiments conducted do not demonstrate reducing the gender bias of the translation model--mitigation is discussed as a motivation but it is not clear that it is actually a major aspect of the work as currently presented.
- The primary novelty of the work is limited to the balancing algorithm provided, as the collection and alignment stages take advantage of existing approaches.
- Evaluation metrics depend heavily on BLEU scores, which may not be sufficiently informative for this kind of work.

---

### Decision · Program_Chairs · 2022-09-16

Accept